# Generalizing to generalize: Humans flexibly switch between compositional and conjunctive structures during reinforcement learning

Nicholas T. Franklin[1,2]*, Michael J. Frank[2,3]*

**1** Department of Psychology, Harvard University, Cambridge, Massachusetts, United States of America,
**2** Department of Cognitive, Linguistic & Psychological Sciences, Brown University, Providence, Rhode Island,
United States of America, **3** Carney Institute for Brain Science, Brown University, Providence, Rhode Island,
United States of America

* nfranklin@fas.harvard.edu (NTF); michael_frank@brown.edu (MJF)

Generalizing to generalize: Humans flexibly switch
between compositional and conjunctive structures
during reinforcement learning. PLoS Comput Biol
16(4): e1007720. https://doi.org/10.1371/journal.
pcbi.1007720

KINGDOM

**Data Availability Statement:** All de-identified data,
analyses and computational model are available in
our GitHub repository, https://github.com/
nicktfranklin/GeneralizingToGeneralize.

## Abstract

Humans routinely face novel environments in which they have to generalize in order to act
adaptively. However, doing so involves the non-trivial challenge of deciding which aspects
of a task domain to generalize. While it is sometimes appropriate to simply re-use a learned
behavior, often adaptive generalization entails recombining distinct components of knowl-
edge acquired across multiple contexts. Theoretical work has suggested a computational
trade-off in which it can be more or less useful to learn and generalize aspects of task struc-
ture jointly or compositionally, depending on previous task statistics, but it is unknown
whether humans modulate their generalization strategy accordingly. Here we develop a
series of navigation tasks that separately manipulate the statistics of goal values ("what to
do") and state transitions ("how to do it") across contexts and assess whether human sub-
jects generalize these task components separately or conjunctively. We find that human
generalization is sensitive to the statistics of the previously experienced task domain, favor-
ing compositional or conjunctive generalization when the task statistics are indicative of
such structures, and a mixture of the two when they are more ambiguous. These results
support a normative "meta-generalization" account and suggests that people not only gener-
alize previous task components but also generalize the statistical structure most likely to
support generalization.

## Author summary

To act in new situations, people not only have to generalize from previous experiences,
but they also have to decide how to do so. One strategy is to re-use behaviors they've
already learned, but this will only be helpful if all aspects of the new situation are similar
enough. Alternatively, people can combine knowledge from multiple sources and devise a
new plan. For example, a skilled musician may re-use the hand motions learned playing

**Funding:** This work was supported in part by the National Science Foundation Proposal 1460604 to MJF, www.nsf.gov. The funders had no role in study design, data collection and analysis, decision to publish, or preparation of the manuscript.

**Competing interests:** The authors have declared that no competing interests exist.

the guitar to play a different style of music on a banjo. Previous theoretical work has suggested that the best strategy is to learn from the statistics of the environment to decide how to best generalize, whereby some environments imply that all parts of a task should be re-used as a whole, whereas others suggest that different components can be generalized separately. Here, we test whether people's generalization strategy changes with their environment using three navigation tasks, in which people have to decide both where they want to go and how to get there. We varied whether it was advantageous to generalize these two pieces of information separately or together and found that people adapted their generalization in line with an optimal computational model of meta generalization. These results suggest that people not only generalize what they learn within a single task, but they also generalize their generalization strategy as well.

## Introduction

It has long been proposed that rather than simply re-use past associations in a novel scenario, humans can flexibly recombine components of prior knowledge to take novel actions [1]. For example, an adept musician can learn multiple instruments by generalizing the motor skills needed to play across instruments, even as they use those skills to different effect across the different instruments. Conversely, they can transfer songs learned on one instrument to another even as the movements needed to play a song on the piano, for example, are very different than that of a guitar. In principle, the sequences of notes used to generate songs are distinct from the skills needed to play an instrument: each is an independent component that can be combined with others arbitrarily. This degree of compositionality is critical for flexible goal-directed behavior but is often lacking in theoretical accounts of human and animal generalization.

Previous models have considered how agents and animals can cluster "latent states" across multiple contexts that share task statistics in both Pavlovian [2] and instrumental learning settings [3, 4]. These models assume each context acts as a pointer to a latent structure, and generalizing task statistics requires inference over which structure the current context belongs to. This form of Bayesian non-parametric clustering and generalization can be approximately implemented in corticostriatal gating networks endowed with hierarchical structure [3] and have been used to explain human generalization behavior and neural correlates thereof in a number of reinforcement learning tasks [3–9].

However, the form of clustering assumed in these models introduces normative challenges that may prevent them from scaling to ecological problems. In these models, task structures are either reused and otherwise learned from scratch, meaning constituent knowledge within each task structure is inseparable. As a consequence, when some observations rule out a given task structure, the agent can no longer generalize any aspects of that structure, thereby preventing the sharing of partial knowledge between contexts, requiring agents to relearn information they already have access to. This all-or-none generalization would, for example, prevent a musician from transferring a song learned on a piano to a guitar, given the differences in required motor actions to produce the desired notes. This form of generalization is representationally greedy, forcing an agent to relearn what it already knows. More problematically, it also tends to be brittle in artificial agents, as policies and policy-dependent representations are often not robust to new tasks [10, 11].

Thus, given the normative challenges posed by generalizing structures as a whole, a key desideratum for clustering models is that they support component-wise generalization, i.e.,

that they can be *compositional*. We note that this desideratum is not limited to clustering models, and there are many possible decompositions of the reinforcement learning problem [12–16]. Often, task components have been derived from a decomposition of the value function over actions and states. Here, we consider a decomposition of the learning problem into the two types of information that determine the value function, information about movement through an environment (the "transition function") and learning information about rewards or goals (the "reward function"). In a reinforcement learning problem, these are are commonly framed as two separate pieces of information about a task combined through planning to determine actions [17]. These two pieces of information are a natural choice for components, as one may have multiple goals (i.e., reward function) in the same environment (i.e., transition function) in different situations or may share the same objectives in distinct environments.

Interestingly, this choice of task components reveals a statistical trade-off in generalization that will drive an adaptive learner to vary its generalization strategy across tasks [18]. Independently generalizing rewards and transitions as task components adds a statistical bias to the generalization that is adaptive when the relationship between the two components across contexts is weak, noisy or difficult to discover. As an analogy, because clustering rewards and transitions independently ignores the relationship between them, it brings a similar set of benefits in limitations as a Naive Bayes classifier. Information will be lost, but this may result in a more robust statistical model [19]. In contrast, joint clustering will, with sufficient experience, learn the correct generalization statistics at a potential cost of sample efficiency. When there is a strong, discoverable relationship between rewards and transitions across contexts, then it is adaptive to generalize them together, as previous models have implicitly assumed [3, 4]. However, the cost of choosing the suboptimal fixed generalization strategy can grow exponentially; a normative agent can circumvent this cost by dynamically arbitrating between these forms of independent and joint clustering as a function of the statistical evidence of each across learning episodes [18]. This "meta-generalization" strategy requires that the agent has access to both joint and compositional task representations and best makes use of them depending on the environment.

It is not well understood how human learners generalize component knowledge in reinforcement learning tasks. However, normative analysis offers testable predictions. If humans learners decompose task structures into rewards and transitions and act adaptively, then we would expect their generalization behavior to vary between a joint and compositional strategy as the statistics of the task environment changes. In the present work, we thus assessed whether human generalization behavior would depend on the extent to which external task statistics are suggestive of independent vs. joint generalization of task structures in three separate experiments that manipulated this hierarchical task statistic.

We developed a novel series of navigation tasks that separately manipulate goal-values ("where do you want to go?"; the reward function) from the actions needed to move in the maze ("how can you get there?"; the transition function over states and actions). Both pieces of information are required to solve the task (i.e., to reach the reward). We manipulated the statistics of these two component-features across contexts and tasks, such that the transition function was more or less informative about the reward function. We then assessed the degree to which humans were able to generalize these learned structures in novel contexts, and whether such generalization was consistent with joint (i.e., entire structure) or independent clustering. To preview our results, we find that subjects vary their generalization strategy with the encountered task statistics, such that they generalize compositional task-components independently when appropriate to do so and jointly when suggested by the task, consistent normative theoretical predictions and a compositional representation of task structure.

## Results

Subjects completed a series of tasks in which they navigated a 6x6 grid-world on a computer in an attempt to discover the reward in one of a set of labeled goal locations across trials (Fig 1). For simplicity, subjects learned a deterministic and uniquely identifiable mapping between arbitrary keyboard presses and movements within the grid-world, as opposed to a complete state-action-state transition function (prior simulations in [18] suggest that learning this reduced action-movement mapping in lieu of a full transition function does not influence the generalization tradeoffs discussed in the current work). These mappings were chosen to be independent, such that it was not possible to learn a mapping on one hand and transfer it to another, either directly or via simple transformation. Similar to the "finger sailing" task [20, 21], this design allows us to study subjects' ability to learn about mappings (state transitions) separately from the goal-values (reward function). Moreover, successful performance in the task requires flexible re-planning on each trial—a form of model-based control [22]: the subjects' initial location and that of the goal were varied from trial to trial, so as to equate the reward value of each button press (i.e., stimulus-response bias). Critically, many of the contexts share the same mapping and/or goal-values, and subjects can boost learning by leveraging this structure [18].

To formally assess alternate learning and generalization strategies in the human navigation tasks, we adapted the computational models previously used to analyze the statistical tradeoff between compositional and joint structure learning [18]. These include a *joint clustering agent*, an *independent clustering agent* and a *meta-generalization agent* that dynamically arbitrates between the two (Materials and methods). All three models are extensions of the joint clustering model proposed by [3] to account for flexible re-use of learned structures across contexts, and make equivalent predictions on the types of instrumental stimulus-response tasks that

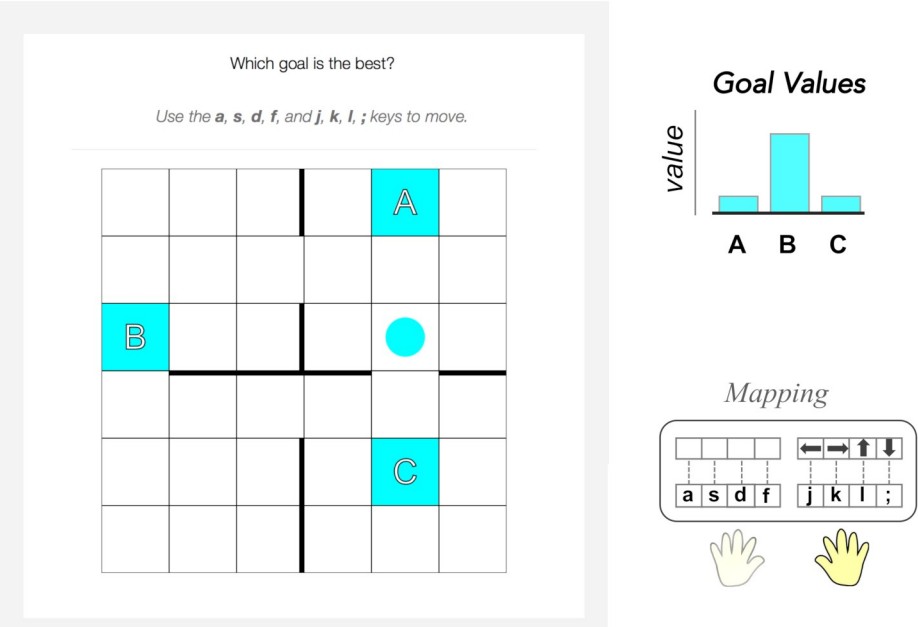

**Fig 1. Subjects controlled a circle agent in a grid world (left) and navigated to one of three potential goals (colored squares labeled "A", "B", or "C").** Context was signaled to the subject with a shared color for the agent and goals. In each context, subjects learned the identity of the rewarded goal within the trial (top right) while also learning a mapping between the keyboard responses and the cardinal movement within the grid world (bottom right).

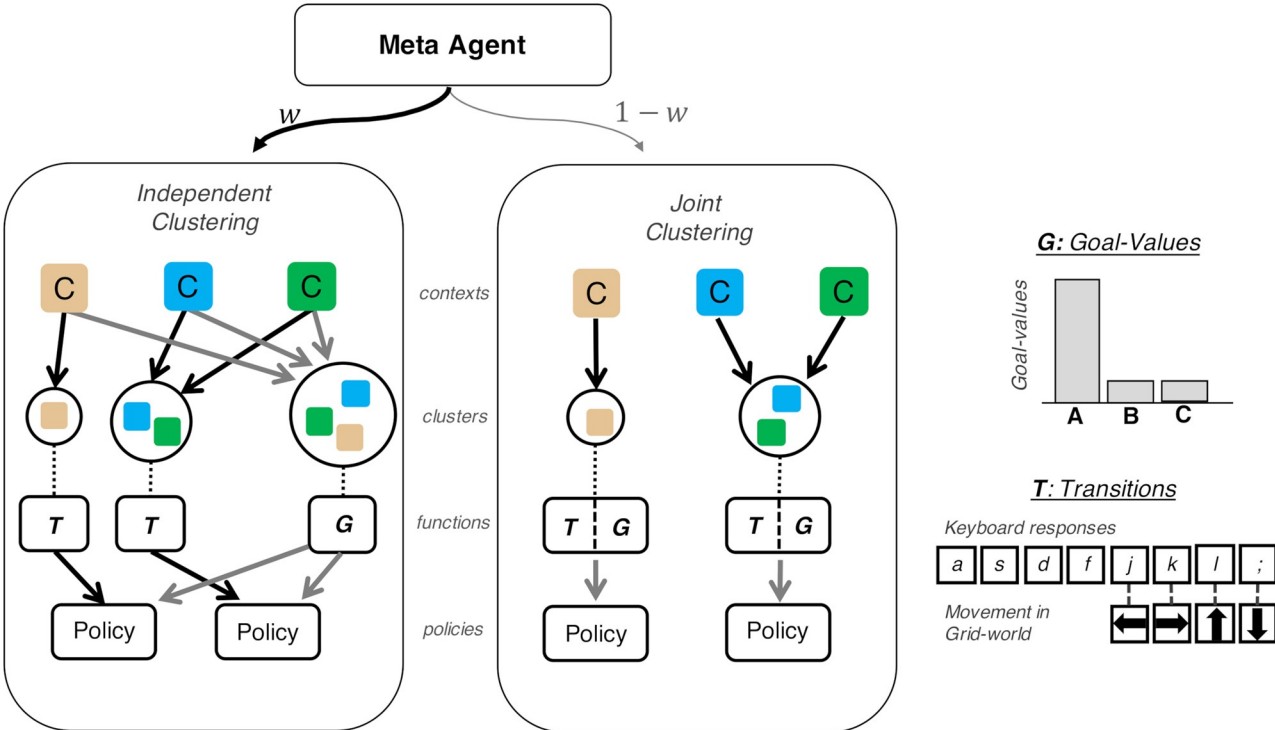

**Fig 2. Schematic depiction of computational agent.** The meta-generalization agent arbitrates between independent clustering and joint clustering according to a learned weight *w*. In both clustering strategies, contexts (colored squares) are grouped into clusters based on the statistics of their associated goal (G), and transition (T) functions. Independent clustering clusters each context twice, once each for goal-value and transition functions, whereas joint clustering assigns each context into a single cluster. Planning derives a behavioral policy from the learned contingencies. Adapted from [18].

have previously been investigated, wherein the reward values were not manipulated separately from the transitions [4, 5]. Each agent probabilistically assigning contexts into clusters that share similar observation statistics via Bayesian inference, with the observation statistics serving as the likelihood and the *Chinese Restaurant Process* (CRP [23]) as a context-popularity based prior. (In these models, "context-popularity" refers to the number of distinct contexts that are assigned to a single cluster, meaning that a contexts that is experienced multiple times only contributes once to context-popularity. This is distinct from a frequency model, which counts each repeated context multiple times.) This context-clustering process allows the agents to reuse previously learned functions in novel contexts, and hence facilitating generalization. Similar mathematical principals underlie prior clustering models of Pavlovian transfer learning [2], category learning [24–26], and memory [27]. Moreover, these agents generalize task structure based on context popularity as a consequence of the CRP prior. In a new context, each agent will reuse task structure as a parametric function of how popular that task structure is across previously encountered contexts.

However, the agents differ by whether they cluster reward and transition functions as separate entities or jointly. The joint clustering agent clusters each context based on the "joint" (conjunctive) statistics of learned state transitions and reward functions (Fig 2). Such an agent, when attempting to generalize learned structures to new contexts, will use information it has gathered about the likely mappings to infer the likely goal values, but as such, it cannot reuse one independently of the other. This is equivalent to generalizing complete policies as indivisible structures. In contrast, the independent clustering agent clusters transitions and rewards

separately by probabilistically assigning each context into two context-clusters, each associated with either a reward or transition function. Such an agent can make inferences about likely goal values that are not tied to any specific mapping, but as such, it cannot improve performance when the two functions are informative about each other. As neither fixed strategy is optimal across unknown task domains, the meta-generalization agent dynamically arbitrates between joint and independent clustering based on the expected value of each (approximately equivalent to the Bayesian model evidence of each), given the experienced task statistics in the environment.

In reinforcement learning problems, information about the transition structure is typically available prior to information about the reward value, and consequently, the three agents differ largely in their generalization of reward functions [18]. While both agents generalize goal values as a function of their popularity across contexts, the joint clustering agent considers this popularity only for the subset of contexts that share the same mapping (in the musician example, this is like inferring what song to play based on its popularity conditioned on the set of instruments that share the same motor mappings). In contrast, the independent clustering agent generalizes goals by pooling across all contexts regardless of mappings. Thus, we can distinguish these model predictions by looking at goal generalization to see whether it varies as a function of mapping. Because the meta-generalization agent arbitrates between these two strategies probabilistically, it predicts a dynamically weighted blend of the two across time.

We exploit this logic in the following grid-world tasks: in each task, subjects learn to navigate to reach a goal in a set of training contexts with varying overlap in mappings and rewards and are then probed for generalization in an unprompted set of novel contexts. Subjects are not told that it is possible to generalize (as it turns out in our designs, it is always advantageous to generalize mappings, but disadvantageous to generalize rewards in all but one of the experimental conditions). Critically, the degree to which mappings were informative of goal values was manipulated across experiments, allowing us to test whether subjects are sensitive to this structure. Across these three experiments, joint and independent clustering each predict a fixed and identifiable strategy, thus allowing us to differentiate between the two on each task qualitatively. The meta-generalization agent, which predicts that subjects adaptively change their generalization strategy to exploit the statistics of the task, varies in its behavior across the three tasks.

To preview the results, we find that subject generalization strategy changes with the meta-statistics of the task and is inconsistent with any single fixed generalization strategy. These results suggest that people rely on multiple strategies to generalize and adapt their behavior to be consistent with the statistics of their environment.

## Experiment 1: Joint structure

In the first experiment, subjects navigated grid-worlds in which there was a consistent relationship between the goal locations and the mappings, such that a given mapping was always paired with a given goal during training (Fig 3, Table 1, S1 Table). This relationship is suggestive of joint structure, and we hence tested whether subjects would later generalize this structure to novel contexts. Prior to an unsignalled generalization probe, subjects completed 32 trials across 3 training contexts, each of which was associated with one of two mappings and one of two potential goals. Subjects were instructed on the relationship between contexts, mappings, and goals during a pre-training instruction but were not told how the relationships generalize between contexts. Subjects received a binary reward (linked to financial payment) for selecting the correct goal and no reward for choosing the other goal. Two of the training contexts shared the same transitions and were paired with the same high popularity goal, while the

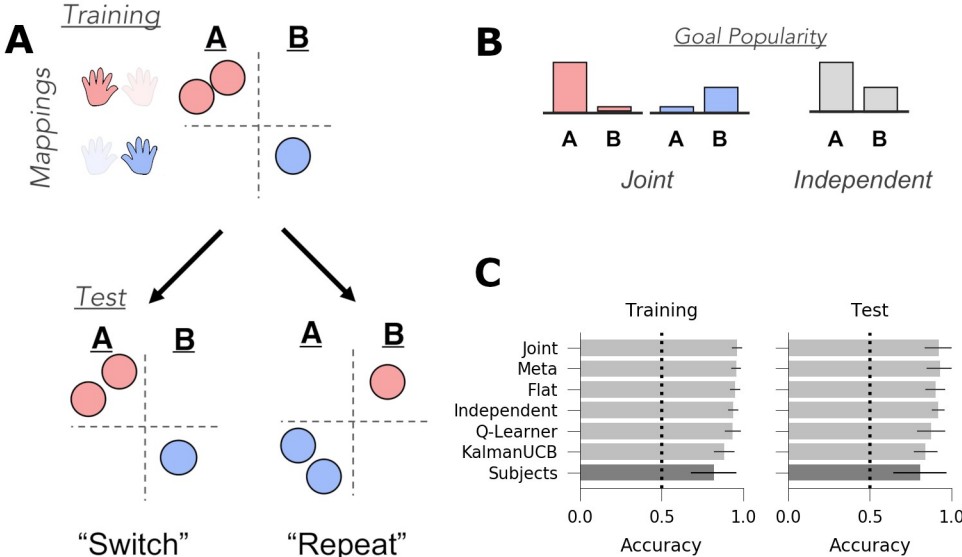

**Fig 3. Experiment 1: Joint structure.** *A*: Task Design: During training, subjects saw three contexts (depicted here as circles) in which a given transition function (mappings, indicated by red/blue hands) was always paired with the same reward function (goals A or B), suggestive of joint structure. In the test phase, subjects saw three novel contexts in which the relationship between transitions and goals was either repeated (i.e., each goal was paired with the same mapping as seen during training) or switched (goal B associated with the mapping that was previously paired with goal A). The switch vs. repeat manipulation was conducted between-subjects. *B*: Goal popularity in the training environment as a function of mapping (as tracked by a joint agent, left) and collapsed across all contexts (as tracked by an independent agent, right). *C*: Goal accuracy of models (light grey) and human subjects (dark grey) across all trials in the training and test contexts. Chance performance denoted with dotted line. Error bars denote standard deviation.

third context was paired with the remaining goal and mapping. Thus, there was a one-to-one relationship between transitions and goals during training. To control for potential stimulus-response biases, the number of trials within each context was balanced such that each mapping and each correct goal was presented in the same number of trials (i.e., the context associated with the low popularity goal/mapping was presented twice as frequently as the other two contexts [4]). In a subsequent test phase, subjects saw three novel contexts in which the joint statistics of the training contexts were either repeated (repeat condition) or switched (switch condition) in a between-subjects manipulation. However, the more popular goal in training (goal "A") remained the more popular goal in testing, regardless of condition.

**Computational modeling.** To confirm the intuition that this task design is indicative of joint structure, we compared the simulated behavior of six computational models on the task,

**Table 1. Test contexts for experiment 1.** Goal popularity, both overall and for contexts with shared mappings, is denoted as the fraction of contexts in training with the same goal.

| Context | Goal | Mapping Popularity | Goal Pop. (Overall) | Goal Pop. (Same Map.) | n Trials |
|---|---|---|---|---|---|
| Repeat Test 1 | A | High | 2/3 | 2/2 | 4 |
| Repeat Test 2 | A | High | 2/3 | 2/2 | 4 |
| Repeat Test 3 | B | Low | 1/3 | 1/1 | 8 |
| Switch Test 1 | A | Low | 2/3 | 0/2 | 4 |
| Switch Test 2 | A | Low | 2/3 | 0/2 | 4 |
| Switch Test 3 | B | High | 1/3 | 0/1 | 8 |

including three generalization agents and three non-generalizing control agents (see Materials and methods). The generalization models were the Joint clustering, Independent clustering and Meta-generalization agents described above [18]. The non-generalizing control agents were a "flat" agent that assigns each context to a new cluster, a Q-learning agent parameterized with a learning rate and an uncertainty-based exploration agent that explores goals based on the upper confidence bound (UCB) of their reward distributions. The key difference between the generalizing and non-generalizing agents is that the generalizing agents pool information across contexts, whereas the non-generalizing agents do not. This process continues throughout all trials, such that the new contexts encountered by the generalizing agents during the test phase can influence subsequent generalization. Note also that all of the generalizing and non-generalizing models learn contextually: the task was designed to require context and aggregating reward across all trials regardless of context leads to chance performance and does not produce any of the qualitative behaviors discussed below (see S4 Text).

Each model was simulated with sampled parameters on 2500 random instantiations of the tasks, and these simulations were sub-sampled to create 200 batches matched to the sample size of human subjects. This allows us to identify a pattern of results that discriminate the predictions of the models across their range of plausible parameter values, marginalizing over the parameters with an analogous logic to a Bayes factor [28] and Bayesian model evidence [29]. This form of simulation naturally penalizes for complexity as models that are too expressive will generate a wide range of possible datasets [29].

Each of the six models successfully learned the task, achieving greater than chance accuracy in both the training and test contexts (Fig 3; $p < 0.005$ for all models). In this task, joint clustering earned the highest reward overall, and because it earned more reward than the flat agent (the top non-generalizing agent) in both the training (accuracy difference $M = 0.02\%$, $p < 0.005$) and test (accuracy difference $M = 0.02\%$, $p < 0.005$), we can conclude that it is adaptive to generalize rewards and transitions jointly in this task.

We confirmed that the independent and joint clustering agents are differentially sensitive to the test context manipulation. In particular, the joint clustering agent predicts better performance in the repeat condition (in which the goals are paired with the same mappings as during training, even in novel contexts) than in the switch condition ($M = 0.194$, 95% highest posterior density interval (HPD) = [0.163, 0.230]), whereas the independent agent predicts no such effects ($M = 0.002$, 95% HPD = [-0.023, 0.026]). Conversely, the independent agent predicts faster learning when the most popular goal is rewarded in either test context (Fig 4; $M = 0.183$, 95% HPD = [0.154, 0.211]), whereas this effect is marginal in the joint agent ($M = 0.023$, 95% HPD = [0.000, 0.056]). (Note that joint clustering *can* be sensitive to goal popularity for a given mapping; see experiment 2). In addition, the meta-generalization agent, which infers which structure is most likely during the training phase, is more similar to the joint agent (effect of test condition: $M = 0.174$, 95% HPD = [0.129, 0.217]) but, like the independent agent, also shows a (much smaller) effect of the rewarded goal ($M = 0.05$, 95% HPD = [0.006, 0.087]). Finally, we confirmed that none of the flat agent, the Q-Learner, nor the upper confidence bound agents were sensitive to these manipulations and showed no differences between the test contexts (Fig 4A and 4C).

**Human behavior.** The behavior of 129 subjects collected online via Amazon Mechanical Turk was carried through to analysis, 49 of which were randomly assigned the switch condition. In both the training and test conditions, subjects were well above chance in goal accuracy (Training: $M = 82.0\%$, one-sided t-test, $t(128) = 26.8$, $p < 10^{-53}$; Test: $M = 80.8\%$; $t(128) = 21.9$, $p < 10^{-44}$). Across training, subject performance improved as a function of time as measured by rewards received, reaction time and navigation efficiency (see S2 Text). Using Bayesian linear modeling, goal accuracy (rewards received) was found to increase as a function of

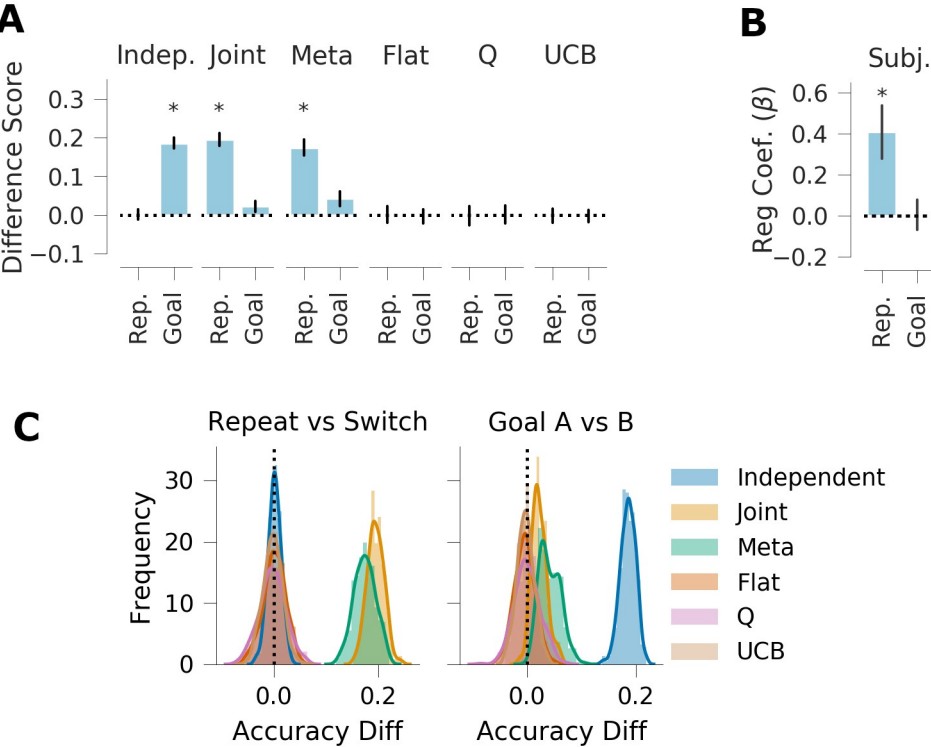

**Fig 4. Generalization performance, experiment 1.** *A*: Qualitative model predictions expressed as difference in goal accuracy between the *Repeat* and *Switch* test conditions (Rep.) and the difference between test contexts with different correct goals (Goal). *B*: Human subject data. Regression weights for the between-subjects comparison of goal accuracy in the two test conditions (Rep.) and the within-subjects comparison of test contexts with different correct goals (Goal). *C*: Histogram of simulated effect sizes in the two comparisons of interest for each of the 6 evaluated models across a range of parameters. Error bars denote standard deviation.

trials within a context (S2A Fig; $\mathbb{E}[\beta_t] = 0.154$, 95% HPD = [0.127, 0.182]) as was their speed (reciprocal reaction time) ($\mathbb{E}[\beta_t] = 0.048$: 95% HPD = [0.002, 0.098]). Navigation (in)efficiency, defined as the number of responses taken in excess of the minimum path length between the start location of the trial and the selected goal, also decreased as a function of trials within each context ($\mathbb{E}[\beta_t] = -0.012$, 95% HPD = [-0.015, -0.008]).

The primary measure of interest was goal accuracy in the test context, which differentiates the predictions of joint and independent clustering models (see above). Goal accuracy was assessed with hierarchical Bayesian logistic regression that included test condition (switch vs. repeat) as a between-subjects measure and correct goal as a within-subjects measure (see Materials and methods). In addition, the number of trials experienced within the same context and whether a trial of the same context was sequentially repeated and previously correct were included as nuisance regressors. Both nuisance regressors significantly predicted test accuracy (times in context: $\mathbb{E}[\beta_t] = 0.34$, 95% HPD = [0.223, 0.470]; sequential correct repeats: $\mathbb{E}[\beta_{rep}] = 1.623$, 95% HPD = [1.002, 2.201]). Consistent with the predictions of joint clustering and the meta-generalization agent, subjects were more accurate in the repeat condition than in the switch condition (Fig 3, $\mathbb{E}[\beta_{repeat}] = 0.41$, 95% HPD = [0.16, 0.66], $p_{1\text{-tail}} < 0.0005$). Accuracy did not vary as a function of the rewarded goal in the test contexts ($\mathbb{E}[\beta_{goal}] = 0.005$, 95% HPD = [-0.15, 0.14]). This pattern of behavior is consistent with the predictions of the joint and meta agents and is inconsistent with the predictions with the independent agent or any non-generalizing agent, suggesting that subjects generalized the task components jointly.

## Experiment 2: Independent structure

While experiment 1 provided evidence for generalization in the task and transfer of component structure jointly, this experiment alone does not suggest subjects would always generalize components jointly in all domains. In experiment 1, there is a strong relationship between mappings and goals in the statistics of the training environment, which normatively favors joint clustering. Nonetheless, the meta-generalization agent (which combines independent and joint clustering and infers which is more likely) also predicted behavior that was similar to the joint agent. The behavior of the meta agent varies with the statistics of the task domain and thus can often produce similar behavior to a single, fixed strategy within a single experiment. Thus, to differentiate a meta-generalization strategy from a single fixed strategy, it is necessary to examine generalization across multiple tasks.

In experiment 2, we wished to test whether subjects would show evidence for independent, compositional, generalization when suggested to by the task environment. Because the meta agent is sensitive to the conditional relationship between goals and mappings, we provided subjects with an environment where this relationship was very weak. Rather than the one-to-one relationship present in experiment 1, the same goal could be re-used with different mappings, and the same mapping could be re-used with different goals, with some goals more popular than others (Fig 5A; Table 2, S2 Table). Subjects completed 112 trials in seven training contexts prior to an unsignaled generalization test with four novel contexts. Each training context was associated with one of two mappings and one binary, deterministically rewarded goal location out of four possible locations.

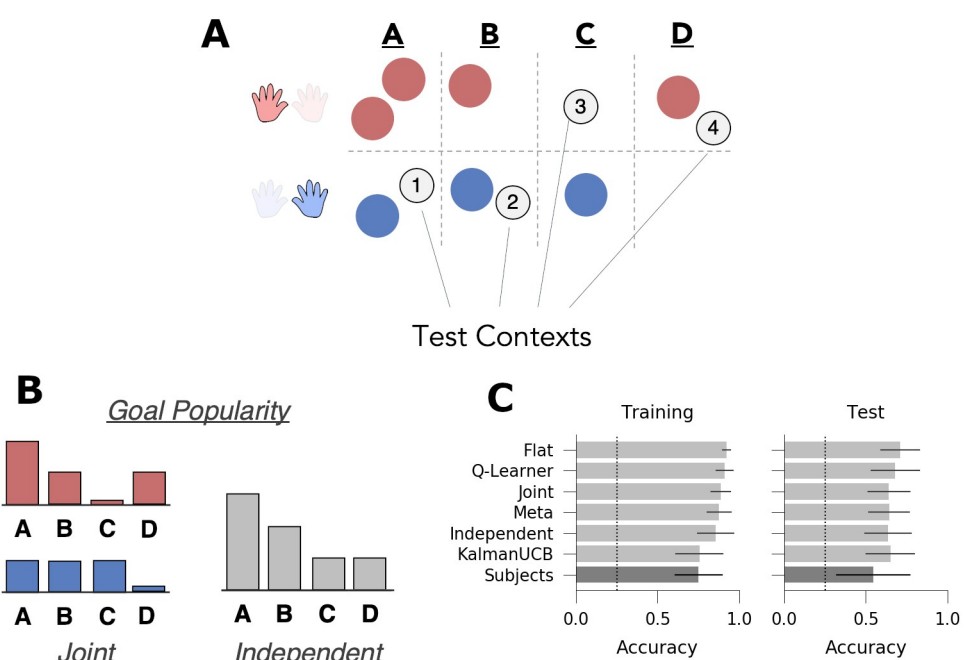

**Fig 5. Experiment 2: Independent structure.** *A*: Subjects saw seven contexts (circles) during training, each paired with one of two mappings (*red*: high popularity, *blue*: low popularity) and one of four goals (A, B, C or D). Two of the high popularity goals were paired with multiple mappings, suggestive of independent structure. Subjects learned to navigate in an additional four novel test contexts following training (grey circles). *B*: Goal popularity in the training environment as a function of mapping (as tracked by a joint agent, left) and collapsed across all contexts (as tracked by an independent agent, right). *C*: Goal accuracy of models (light grey) and human subjects (dark grey) across all trials in the training and test contexts. Chance performance denoted with dotted line. Error bars denote standard deviation.

**Table 2. Test contexts for experiment 2.** Goal popularity, both overall and for contexts with shared mappings, is denoted as the fraction of contexts in training with the same goal.

| Context | Goal | Mapping Popularity | Goal Pop. (Overall) | Goal Pop. (Same Map.) | n Trials |
|---------|------|--------------------|--------------------|-----------------------|----------|
| Test 1  | A    | Low                | 3/7                | 1/3                   | 6        |
| Test 2  | B    | Low                | 2/7                | 1/3                   | 6        |
| Test 3  | C    | High               | 1/7                | 0/4                   | 6        |
| Test 4  | D    | High               | 1/7                | 1/4                   | 6        |

Note that, as before, independent clustering predicts that subjects learn about the statistics of goals and mappings independent of the other, and as such, they would learn about the popularity of goals across contexts regardless of mapping. For example, goal A is the most popular goal overall (marginalizing over mappings) but is equally popular as other goals within the contexts paired with the low popularity mapping (Fig 5B). Thus, independent and joint clustering make qualitatively different predictions about which goals will be searched first in a novel context as a function of the associated mapping. As such, subjects were given four novel contexts in an unsignaled test phase chosen to differentiate joint and independent clustering by crossing the *overall* goal popularity with the *conditional* goal popularity (conditioned on the mapping; Fig 5A and 5B; Table 2). If subjects learn independent structure, then they should more rapidly learn for novel contexts associated with the more popular goal A (i.e., test context 1) even if it is paired with the low popularity mapping, whereas joint clustering predicts no such advantage.

This same logic sets up within-subject performance comparisons between pairs of test contexts. Here, we focus on performance conditioned on a single mapping as these are the most relevant comparisons. Thus, we compare goal-accuracy in test context 1 to 2 and test context 3 to 4. In the first comparison, test contexts 1 and 2 have the same conditional goal popularity but different overall popularity. In the second comparison, test contexts 3 and 4 have the same overall popularity but different conditional popularity. Thus, an agent that generalizes based on the overall popularity will show a difference between test contexts 1 and 2 but not between contexts 3 and 4, whereas an agent that generalizes based on the conditional popularity will show the opposite result.

**Computational modeling.** We sought to test whether this task design was indicative of independent rather than joint structure. Each of the six computational models was simulated on the task to generate predictions in these contexts (see Materials and methods). All of the models successfully learned the task, with above-chance accuracy in both the training and test contexts (Fig 5C; all p's<0.005). Importantly, because of the nature of the test contexts chosen to differentiate the models, none of the generalizing agents achieved greater reward than the non-generalizing flat model that learns anew for each context (all p's $<10^{-5}$ in favor of the flat model over clustering agents in both training and test), suggesting the task design did not encourage subjects to generalize, even though other situations reveal substantial and compounding advantages for these generalization models [18]. Thus, we can interpret any evidence for generalization in this task as spontaneous as the task environment does not incentive generalization [3].

The key manipulations of interest are expressed as a difference score in the goal accuracy between test contexts 1 and 2, and the difference between test contexts 3 and 4. As previously noted, test context 1 is associated with goal "A", which is the most popular goal overall. Because test context 2 is associated with goal "B", which has lower overall popularity, independent clustering predicts higher average reward in test context 1 relative to test context 2 (Fig 6;

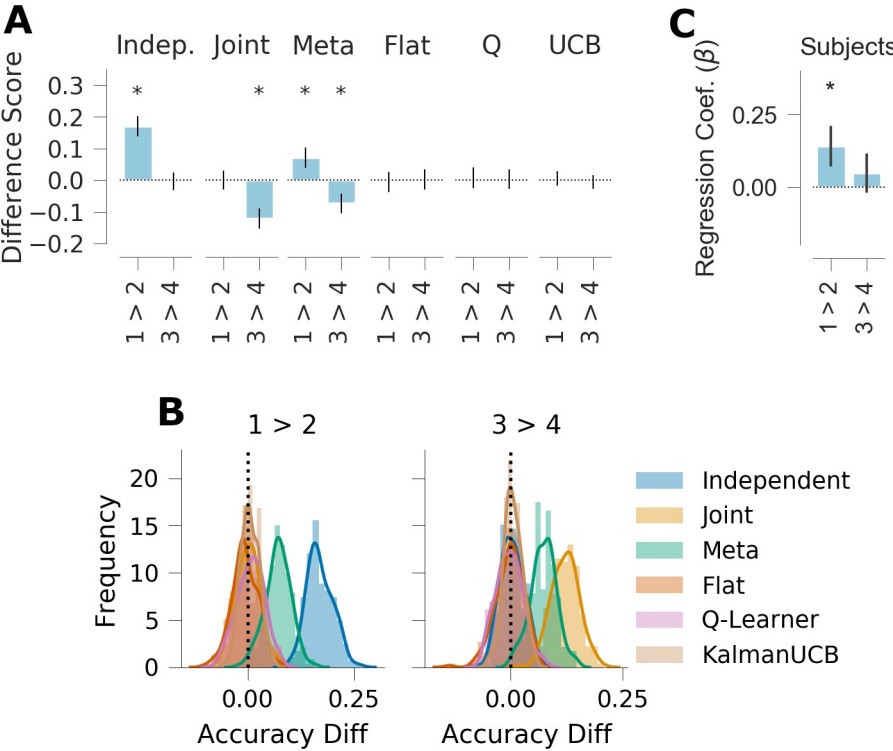

**Fig 6. Generalization performance, experiment 2.** *A*: Qualitative model predictions expressed as the difference scores *Context 1—Context 2* and *Context 3—Context 4*. *B*: Human subject data. Regression weights for the two within-subjects comparisons. *C*: Histogram of simulated effect sizes in the two comparisons of interest for each of the 6 evaluated models across a range of parameters. Error bars denote standard deviation.

M = 0.165, 95% HPD = [0.104, 0.237]). Joint clustering predicts no difference between these two conditions (M = -0.001, 95% HPD = [-0.051, 0.052]) because goals "A" and "B" have the same popularity conditioned on the mapping. Conversely, independent clustering predicts no difference between test contexts 3 and 4 because the associated goals have the same overall popularity (M = -0.003, 95% HPD = [-0.056, 0.048]), whereas joint clustering predicts a negative difference due to the different conditional popularity (M = -0.125, 95% HPD = [-0.186, 0.058]).

As the meta-generalization agent probabilistically weighs these two strategies according to their evidence, and because the training environment does not rule out some joint structure altogether (i.e., some goals are experienced multiple times with one mapping), it shows both effects (1 vs. 2: M = 0.070, 95% HPD = [0.015, 0.126]; 3 vs. 4: M = -0.071, 95% HPD = [-0.141, -0.0146]). Importantly, none of the three non-generalizing agents predict a difference between these two contexts, suggesting that any difference in these metrics can be interpreted as a measure of generalization.

For completeness, we also compared goal-accuracy between the two mappings (i.e., test context 1 and 2 vs. test context 3 and 4), affording three orthogonal contrasts for the regression model. All three generalization agents earned more reward in the low popularity mapping (test contexts 1 and 2) then in the high popularity mapping (test contexts 3 and 4; independent: M = 0.256, 95% HPD = [0.208, 0.300], joint: M = 0.161, 95% HPD = [0.118, 0.207], meta: M = 0.234, 95% HPD = [0.189, 0.278]). Both conditioned on the mapping and overall, the goals associated with test contexts 1 and 2 were more popular than those associated with test

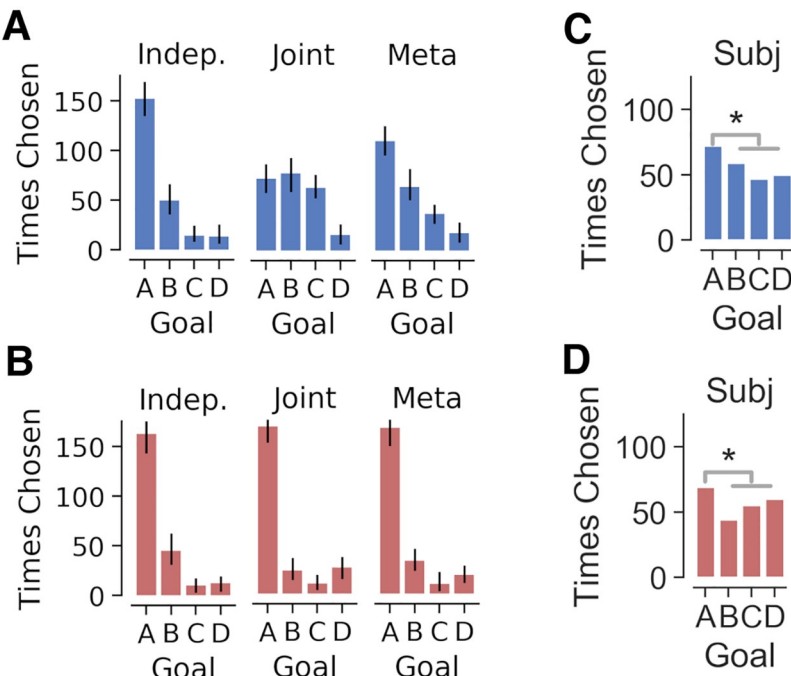

**Fig 7. Goal selection in the first trial of novel test contexts in experiment 2, separated by associated mapping transitions (blue and red).** *A*, *B* Number of times each goal was selected by the independent, joint and meta computational agents in test contexts 1 & 2 (*A*), which shared the low-popularity transition (mapping) function, and in test contexts 3 & 4 (*B*), which shared the high-popularity transition function. *C*, *D* Like the independent agent, subjects chose goal A more frequently than the other goals across both contexts. Error bars denote standard deviation.

contexts 3 and 4. As expected, none of the flat agents showed a difference between these contexts.

Beyond average reward per context, the generalizing agents are further differentiated by the goals selected in the first trial of a new context (Fig 7A and 7B). Independent clustering predicts that goals will be chosen in these trials with the most popular goal selected most frequently, regardless of the associated mapping. In contrast, both joint clustering and meta-generalization predict the goal selection in these trials will be influenced by the mapping. This leads to the qualitative prediction that in the first trial of test contexts 1 and 2, independent clustering and the meta agent predict subjects will select goal A more often than either goal B or C, while joint clustering does not make this prediction. All the non-generalizing agents explore the first goal in a novel context uniformly, regardless of mapping. This occurs because, without information from previous contexts, there is no reason to prefer one goal over another.

**Human behavior.** We analyzed the behavior of 114 subjects in experiment 3. As in the previous experiments, subjects were well above chance performance (25%) when selecting goals in both the training and the test contexts (Fig 5C; Training: M = 75.1%, one-sided t-test, $t(113) = 37.2, p < 10^{-64}$; Test: M = 54.7%; $t(113) = 14.2, p < 10^{-26}$). Overall performance was further assessed with goal accuracy, reciprocal reaction time and navigation efficiency, all of which improved as a function of the number of trials within the training contexts (see S2 Text). The number of trials per context was found to predict accuracy (S3A Fig; $\mathbb{E}[\beta_t] = 0.115$, 95% HPD = [0.108, 0.121]), reciprocal reaction time ($\mathbb{E}[\beta_t] = 0.016$: 95% HPD = [0.014, 0.017]) and navigation efficiency ($\mathbb{E}[\beta_t] = -0.006$, 95% HPD = [-0.007, -0.005], indicating subject performance improved over the course of training.

Analysis of the *a priori* test context contrasts was consistent with the predictions of independent clustering and meta-generalization. These contrasts were assessed with a hierarchical Bayesian logistic regression that included time with context, a subject-specific bias term, and whether a trial of the same context sequentially followed a correct trial as nuisance variables (see Materials and methods). Test accuracy was predicted by both nuisance factors and increased with the number of times a context was seen ($\mathbb{E}[\beta_t] = 0.48$, 95% HPD = [0.413, 0.547]) and when a trial of the same context was repeated sequentially and previously correct ($\mathbb{E}[\beta_{rep}] = 0.921$, 95% HPD = [0.668, 1.155]).

Of the two contrasts of interest, the difference between contexts 1 and 2 was statistically significant, (Fig 6D; $\mathbb{E}[\beta_{1>2}] = 0.140$, 95% HPD = [0.012, 0.266], $p_{1\text{-tail}} = 0.0135$). Importantly, a positive value of this contrast was predicted by the independent clustering and meta-generalization agents, and no other model. The second contrast of interest, the difference between test contexts 3 and 4 was not significantly different from zero. ($\mathbb{E}[\beta_{3>4}] = 0.0476$, 95% HPD = [-0.069, 0.174]). A negative value of this value was predictive by joint clustering and meta-generalization and no other model. For completeness, we also examined the contrast between the two mappings (i.e., test contexts 1 and 2 vs. 3 and 4), an effect which was predicted to be positive by all three generalizing strategies. We did not find evidence of this effect in our subject pool ($\mathbb{E}[\beta_{1\&2>3\&4}] = 0.060$, 95% HPD = [-0.031, 0.144], $p_{1\text{-tail}} = 0.097$).

Follow-up analyses suggested these effects were driven by positive transfer in test context 1. Accuracy was higher in test context 1 than 3 and 4 (context 1 vs. 3 & 4: $\mathbb{E}[\beta_{1>2} + \beta_{1\&2>3\&4}] = 0.199$, 95% HPD = [0.045, 0.364]) but there was no difference between accuracy in test context 2 than in 3 and 4 (context 2 vs. 3 & 4: $\mathbb{E}[\beta_{1\&2>3\&4} - \beta_{1>2}] = -0.080$, 95% HPD = [-0.230, 0.087]). Thus, accuracy in test context 1, which was associated with the most popular goal overall, was higher than in the other three contexts whereas we did not find a difference between the remaining test contexts. These results are consistent with the generalization of the most popular goal overall (goal A) but not the parametric effects predicted by the models.

Subjects' goal selection in the first trial of a test context was also consistent with independent clustering and meta-generalization but not joint clustering. Goal choice was analyzed at a group level with two binomial models (Materials and methods). Consistent with the predictions of independent clustering and meta-generalization, across all trials the choice-probability of goal A was above chance ($\mathbb{E}[\theta_A] = 0.309$, 95% HPD = [0.268, 0.353]) and higher than the average choice probability of goals B and C ($\mathbb{E}[\theta_A - \frac{1}{2}(\theta_B + \theta_C)] = 0.085$, 95% HPD = [0.034, 0.135]). Furthermore, this was true regardless of the associated mapping. Choice probability for goal A was greater than the average goal probability for goals B and C in contexts 1 and 2 (Fig 6C; $\mathbb{E}[\theta_A - \frac{1}{2}(\theta_B + \theta_C)] = 0.083$, 95% HPD = [0.009, 0.150]) and in test contexts 3 and 4 (Fig 6D; $\mathbb{E}[\theta_A - \frac{1}{2}(\theta_B + \theta_C)] = 0.087$, 95% HPD = [0.017, 0.157]).

Thus, we find evidence that subjects generalized the goal with the highest overall context popularity, regardless of the mapping presented in the context. We did not find evidence that goal generalization was parametrically proportional to context-popularity (e.g., preference for goal B over goal C or D), a prediction of the three generalization models as a consequence of the CRP prior.

## Experiment 3: Ambiguous structure

Experiments 1 and 2 showed that subjects generalization learning in novel contexts was more similar to a joint agent when the statistics of the training environment supported joint structure (experiment 1) and more like an independent agent when the statistics supported independent structure (experiment 2). In experiment 3, we provided subjects with a task

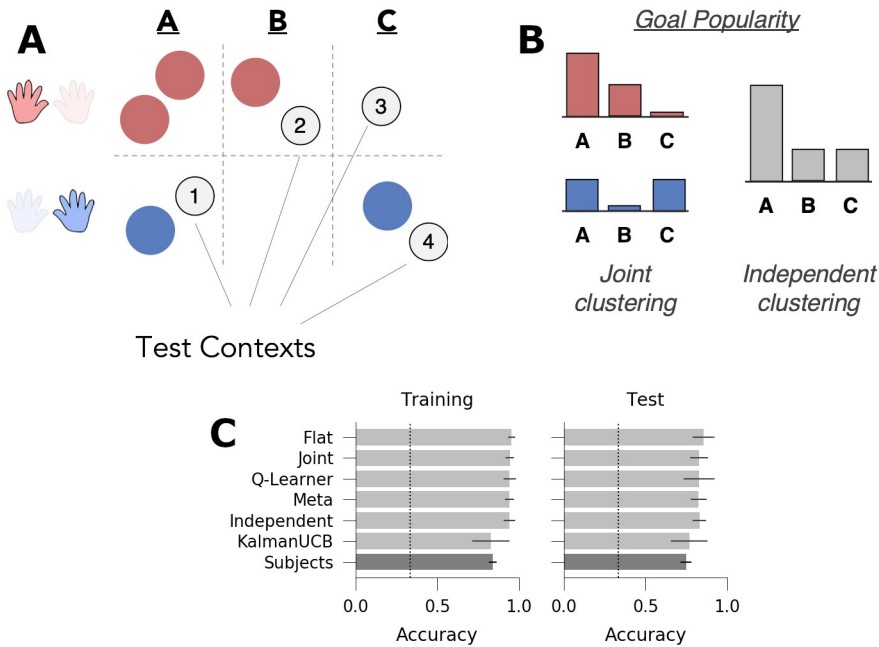

**Fig 8. Experiment 3: Ambiguous structure.** *A*: Subjects saw five contexts (circles) during training, each paired with one of two mappings (*red*: high popularity, *blue*: low popularity) and one of three goals (A, B, or C). The most popular goal A was paired with multiple mappings, but the other goals B and C were paired with a single mapping each, making the structure more ambiguous. Subjects learned to navigate in an additional four novel test contexts following training (grey circles). *B*: Goal popularity in the training environment as a function of mapping (as tracked by the joint agent, left) and collapsed across all contexts (as tracked by the independent agent, right). *C*: Goal accuracy of models (light grey) and human subjects (dark grey) across all trials in the training and test contexts. Chance performance denoted with dotted line. Error bars denote standard deviation.

environment with a more ambiguous relationship between mappings and goals in order to probe whether subjects would show evidence of both joint and independent clustering within the same task. This mixture of joint and independent clustering is a key prediction of the meta-generalization agent, which is able to consider both forms of structure and to make inferences in novel environments that are influenced by the prior training statistics. Indeed, across both previous experiments, subjects behave more similarly to the meta-generalization agent than either joint or independent clustering alone, but we have not yet provided evidence for meta-generalization within a single task.

Thus, in the final experiment, we presented subjects with an environment with a more ambiguous relationship between goal-values and state-transitions across contexts (Fig 8). In this case, the joint vs. independent structure statistics were more ambiguous: goal A was the most popular and paired with different mappings (consistent with independent structure), whereas goals B and C were always paired with a single mapping (consistent with joint structure). Furthermore, goal A was the most popular goal overall but was equally popular to goal C conditional on experiencing the lower-popular mapping. As in the prior experiment, this design differentiates the predictions of independent and joint clustering, as the former is sensitive to the overall goal popularity whereas the latter is sensitive to the popularity conditioned on the associated transition function. Subjects completed 120 training trials across five contexts prior to completing 30 trials in four novel test contexts chosen to probe generalization (Fig 8A; Table 3, S3 Table).

This task has a more ambiguous relationship between mappings and goals than either experiment 1 or experiment 2. This is most clear in the mutual information between mappings

**Table 3. Test contexts for experiment 3.** Goal popularity, both overall and for contexts with shared mappings, is denoted as the fraction of contexts in training with the same goal.

| Context | Goal | Mapping Popularity | Goal Pop. (Overall) | Goal Pop. (Same Map.) | n Trials |
|---------|------|--------------------|--------------------|-----------------------|----------|
| Test 1 | A | Low | 3/5 | 1/2 | 10 |
| Test 2 | B | High | 1/5 | 1/3 | 10 |
| Test 3 | C | High | 1/5 | 0/3 | 5 |
| Test 4 | C | Low | 1/5 | 1/2 | 5 |

and goals within the training contexts, which we normalize here by overall goal entropy for the purpose of cross-experiment comparison (see S3 Text). When normalized mutual information (NMI) is 1, there is a perfect correspondence between mappings and goals and when NMI is 0, there is no relationship. When we evaluate the training sets of each of the experiments, we see a correspondence between NMI and generalization strategy. In experiment 1, where we found evidence for joint clustering, the relationship between mappings and goals is strongest (NMI = 1.0). In experiment 2, where we found evidence for independent clustering, the relationship was the weakest (NMI $\approx$ 0.16). In experiment 3 the relationship between mappings and goals is somewhere between the other two experiments (NMI $\approx$ 0.3). As updating within the meta-generalization agent is sensitive to the correspondence between mappings and goals [18], we would thus expect more of a blend of joint and independent clustering in experiment 3.

**Computational modeling.** As in experiment 2, there was no incentive to generalize, as confirmed by simulations showing that none of the three generalizing agents accumulated greater total reward than the non-generalizing flat agent during either the training or test contexts (Fig 8; all p's<.005 in favor of the flat model). We thus again designed the task such that it was possible to discriminate between predictions of different types of generalization models without incentivizing generalization. Two comparisons, the contrast between test context 1 and 4 and the contrast between contexts 2 and 3, differentiate the predictions of joint and independent clustering. Test contexts 1 and 4 shared the low-popularity mapping and were paired with goals A and C, respectively. Goal A was the highest popularity goal overall but was equally popular as goal C when conditioned on the low-popularity mapping. Thus, the independent agent performs better in test context 1 compared to test context 4 (Fig 9A; M = 0.265, 95% HPD = [0.228, 0.296]), whereas the joint agent shows no such advantage (M = 0.12, 95% HPD = [-0.023, 0.047]). Conversely, consider the contrast between test contexts 2 and 3, both of which were paired with the high-popularity mapping, but were associated with goals B and C, respectively. These two goals shared the same overall popularity but differed in popularity conditional on the high popularity mapping. As such, the joint clustering agent performs better in test context 2 than 3 (M = 0.148, 95% HPD = [0.112, 0.195])), while independent clustering shows no difference (M = -0.005, 95% HPD = [-0.037, 0.025]). Critically, the meta-generalization agent showed patterns consistent with both independent clustering (test context 1 > 4: M = 0.142, 95% HPD = [0.098, 0.184]) and joint clustering (test context 2 > 3; M = 0.058, 95% HPD = [0.023, 0.088]). For the purpose of completion, we also compare goal accuracy between the two mappings (contexts 1 and 4 vs. 2 and 3), providing the third orthogonal contrast for the regression model; all three models predicted a positive value for this difference (independent: M = 0.137, 95% HPD = [0.117, 0.158]; joint: M = 0.131, 95% HPD = [0.103, 0.158]; meta: M = 0.129, 95% HPD = [0.103, 0.160]). As in the prior two experiments, neither the flat agent, the Q-learning agent nor the upper confidence bound exploration agent showed any difference in these contrasts, confirming these metrics as measures of generalization.

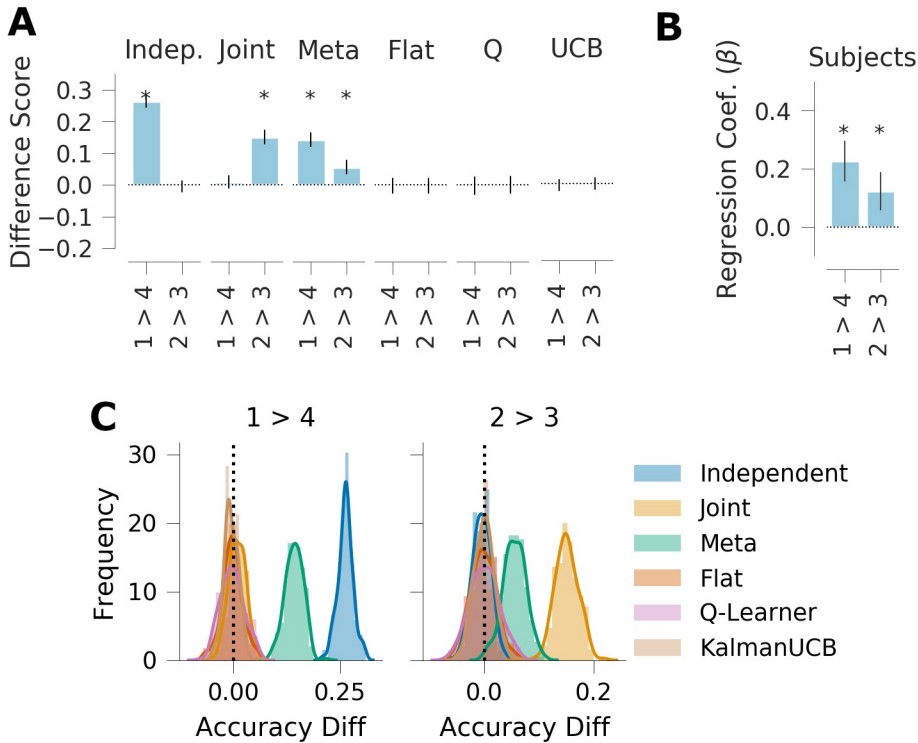

**Fig 9. Generalization performance in experiment 3.** *A*: Qualitative model predictions expressed as the difference scores *Context 1—Context 4* and *Context 2—Context 3*. *B*: Human subject data. Regression weights for the two within-subjects comparisons. *C*: Histogram of simulated effect sizes in the two comparisons of interest for each of the 6 evaluated models. Error bars denote standard deviation.

In sum, the three models make qualitatively different predictions. The joint model predicts a cost in test context 3; the independent model predicts an advantage in test context 1, and, given the ambiguity of the statistical structure during training, meta-generalization predicts a mixture of both effects (Figs 8 and 9).

**Human behavior.** The behavior of 115 subjects was analyzed on the task. As for prior experiments, subjects successfully learned the task and chose goals well above chance in both the training and test contexts (Fig 8C; Training: M = 83.2%, one-sided t-test, t(114) = 53.3, $p < 10^{-81}$; Test: M = 65.8%; t(113) = 20.9, $p < 10^{-40}$). Goal accuracy, reciprocal reaction time and navigation efficiency were assessed as a function of time using Bayesian linear modeling (see S2 Text). The number of trials per context was found to predict accuracy (S4 Fig; $\mathbb{E}[\beta_t] = 0.109$, 95% HPD = [0.098, 0.012]), reciprocal reaction time ($\mathbb{E}[\beta_t] = 0.008$, 95% HPD = [0.0063, 0.0091]) and navigation efficiency ($\mathbb{E}[\beta_t] = -0.003$, 95% HPD = [-0.0043, -0.0027]), indicating subject performance improved over the course of training.

Accuracy in the test contexts provided support for the meta-learning agent over either the joint or independent clustering agent (Fig 9B). Test context accuracy was assessed with hierarchical Bayesian logistic regression where the *a priori* model predictions were instantiated as contrasts between contexts (1 vs. 4 and 2 vs. 3), with the number of trials per context, and whether a context was sequentially repeated and previously correct were included as nuisance regressors (Materials and methods). Subjects' accuracy increased with the number of trials within each test context ($\mathbb{E}[\beta_t] = 0.431$; $p_{1\text{-tail}} < 0.0005$, 95% HPD = [0.352, 0.501]) and when a context was repeated on the next trial and previously correct ($\mathbb{E}[\beta_{rep}] = 0.752$; $p_{1\text{-tail}} < 0.0005$, 95% HPD = [0.486, 1.019]). Critically, subjects were more accurate in test context 1

than 4 ($\mathbb{E}[\beta_{1>4}] = 0.231$, 95% HPD = [0.098, 0.370], $p_{1\text{-tail}} < 0.0005$) and were more accurate in test context 2 than in 3 ($\mathbb{E}[\beta_{2>3}] = 0.123$, 95% HPD = [0.003, 0.253], $p_{1\text{-tail}} < 0.0275$). For completion, we also examined the difference in goal accuracy between the two mappings (i.e., test contexts 1 and 2 vs. 3 and 4). Consistent with all three generalization agents, subjects were also more accurate in contexts 1 and 4 than in 2 and 3 ($\mathbb{E}[\beta_{1+4>2+3}] = 0.101$, 95% HPD = [0.014, 0.194], $p_{1\text{-tail}} < 0.0125$).

As a follow-up analysis, we further examined positive and negative transfer in the first trial of a test context (above or below chance performance on the first trial). Both independent clustering and the meta-generalization agent predict positive transfer in test context 1 and negative transfer in the other test contexts, given the overall popularity of goal A during training. Joint clustering does not predict positive transfer in any context and predicts negative transfer in contexts 1, 2 and 3. Consistent with independent clustering we found positive transfer in test context 1 ($y \sim Binom(\theta)$; $\Pr\left(\theta_1 > \frac{1}{3}\right) = 0.963$, 95% HPD = [0.325, 0.500]) but did not find evidence of positive or negative transfer in the other test contexts ($y \sim Binom(\theta)$; $\theta_2$: 95% HPD = [0.29, 0.46]; $\theta_3$: 95% HPD = [0.23, 0.40]; $\theta_4$: 95% HPD = [0.21, 0.38]).

Overall, these results are consistent with the predictions of meta-generalization and are not fully captured by either the independent or joint clustering.

## Discussion

In the current work, we evaluated human generalization against the predictions of the three, dissociable generalization strategies proposed in [18]. The independent clustering model assumes that subjects generalize in novel environments based on the overall popularity of the previous goal and mappings independently. The joint clustering model assumes that subjects re-visit goal locations in proportion to how often they were paired with each mapping. Finally, meta-generalization assumes that subjects learn the overall statistical relationship between goals and mappings and then generalize based on the evidence for independent or joint structure. Across the suite of tasks, we provided evidence that humans vary their generalization strategy with the learned statistics of their environment in an adaptive way. This suggests that humans leverage compositional representations when generalizing in reinforcement learning tasks and adaptively respond to the statistical challenges of generalization.

In the process of doing so, we also replicated prior work demonstrating that human subjects exhibit positive transfer for previously learned task-sets (stimulus-response-outcome relationships) [3, 5], with increasing generalization performance for those rules that had been most popular across training contexts [4]. Even when popular contexts were experienced fewer times, subjects tended to generalize the goals associated with popular contexts, suggesting context popularity-based generalization, rather than raw frequency. Further, subjects generalized in experiments 2 and 3 even as it was disadvantageous, replicating the prior finding suggesting that humans will generalize spontaneously even if it requires paying a cost to do so [3, 6].

However, prior empirical work assumed that task-sets are generalized as a whole, and thus amount to joint clustering. The current set of studies was designed to disrupt and manipulate the relationship between transition and reward functions. Consistent with normative accounts [18], subjects were able to generalize each function independently of the other, particularly when the training environment was suggestive of such independent structure. These results provide evidence for the flexibility of human generalization, including the tendency to recombine component pieces of knowledge that have not been experienced together, to new contexts. For example, in experiments 2 and 3 subjects tended to generalize the most popular goal overall, regardless of the mappings associated with it during training. This situation is analogous to a musician that can transfer popular songs learned across multiple instruments to

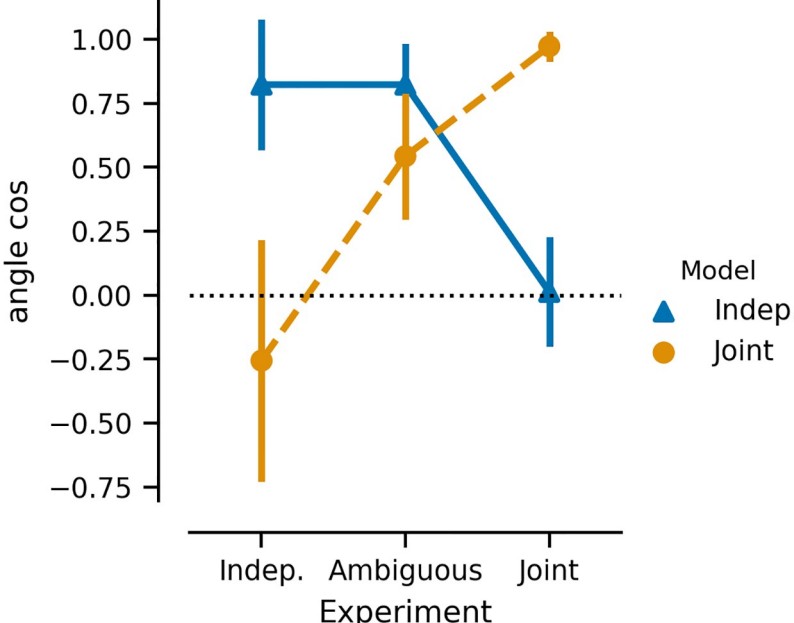

**Fig 10. Comparison of human behavior to model to model predictions.** Angle cosign between the vector of *a priori* model contrasts and human subject regression coefficients is shown for independent clustering (blue) and joint clustering (orange) across experiments 1 (joint structure), 2 (independent structure), and 3 (ambiguous structure) ordered by degree of joint structure in the tasks. Positive values (max = 1) reflect similarity between model prediction and human behavior.

another instrument even with very different physical actions needed. However, when the task statistics implied joint structure, subjects were able to harness that relationship to generalize accordingly, repeating goals that had been previously paired with the same mapping over those that had been reached using a different mapping function (experiment 1 and partially experiment 3). This situation is analogous to exhibiting a preferential tendency to play a song on a new instrument that has shared actions required to produce the desired effect.

Why do we expect these tasks to produce these generalization strategies? If subjects are acting adaptively, then we would expect them to use the statistics of the training set to inform the nature of their generalization strategy on the test set, thus amounting to a form of meta-generalization. This meta-generalization is normatively driven by the degree to which state transitions and rewards are mutually informative and the pattern of human generalization behavior is similarly governed. In our three experiments, we observe that subject behavior is most similar to joint clustering in experiment 1, where the mutual information is high, and most similar to independent clustering in experiment 3, where the mutual information is lowest (Fig 10). Thus, the degree to which each fixed strategy accounts for human behavior depends on the task statistics as meta-generalization predicts.

Previous empirical work has provided support for clustering models as an account for generalization of both operant [4, 5] and Pavlovian values [30, 31]. This approach to generalization is qualitatively similar to those found in the machine learning literature that rely on policy reuse in multi-task settings [32–34]. Alternatively, an agent may make inferences about unseen portions of the task space solely relying on information present in the task, either through clustering portions of the task space [35, 36] or by function approximation [37]. Humans appear to both cluster the task state for the purpose of planning [38] and interpolate rewards

continuously in unvisited states [39], and moreover, generalize within and across tasks simultaneously [8].

Mechanistically, learning about latent task-set structure is thought to involve the same, albeit hierarchically nested, frontostriatal circuits that are responsible for simple reward feedback-based learning [3–5]. How might this circuitry adapt to assume joint vs. independent clustering? One simple mechanism to enforce compositional representations of goals and transitions would be to embed them in independent systems. For example, learning motor mappings might involve interactions among basal ganglia, cerebellum and/or motor areas of the cortex [20, 21, 40], while learning reward/goal values may involve interactions among prefrontal areas representing value and/or state [41–44]. Thus the ventral striatal orbitofrontal circuit could learn goal or state values that can be reused independently of the actions needed to reach them [42, 45], thus potentially facilitating compositional recombination.

In contrast, when higher-order rules or goal values are used to constrain physical mappings, such joint structure can be learned in hierarchical rostrocaudal circuits that embed multiple levels of abstraction [3, 7, 46]. The mechanisms by which the brain could infer which of these structures (orthogonal/independent vs. hierarchical) should be used remain underexplored. Nevertheless, previous modeling and empirical work has suggested that reinforcement learning principles can be used to engage the level of rostrocaudal hierarchy needed for a given task, and when such hierarchy is not present, reward prediction errors are used to prevent gating of these circuits in favor of combinatorial representations [7, 46]. Indeed, these studies found that the degree of evidence of hierarchical structure using a Bayesian mixture of experts, akin to our meta-generalization agent, was related to the development of hierarchical gating policy in the neural network [3, 46]. More recent work has suggested this form of structure learning is neurally dissociable from the associative learning and involves the rostrolateral prefrontal cortex and angular gyrus [9].

## Limitations

The experiments presented here were designed to show the effects of joint and independent clustering on goal generalization during a defined generalization phase and did not examine other aspects of generalization, including generalization during training, interactions with planning, and mapping generalization. Detecting generalization during training can be difficult, as subjects have to simultaneously learn task structure and associative contingencies. Previous related work has found clear evidence for generalization effects only after sufficient training experience was accrued to infer the most likely structure that affords context clustering [4]. In the experiments presented above, joint and independent clustering are not distinguishable during training in the sample sizes we have collected.

A second limitation is that in each of the experimental tasks, the mappings subjects learned to navigate the mazes represent a minimal version of a transition structure needed to the predictions of the model. This was done to be amenable to learning while still presenting a learning challenge. Similarly, the planning problem itself, in terms of cardinal movements (but not button responses) in the maze was simple. While prior simulations suggest that the type of goal-generalization effects we are primarily concerned with here is not affected by this minimal version of a transition function [18], we do not know if this holds for human learners in more complex domains. Moreover, planning is computationally expensive and poorly understood. It has been shown in purely computational agents that generalize parameterized skills (sequential policies) can reduce exploration costs in novel domains [47], but whether humans generalize similarly is unknown. A particularly interesting proposal is that a composition of multiple, simultaneous actions using a linear approximation to the Bellman equation can result in rapid

generalization to a change in rewards [16, 48]. These linear approximations have recently been argued to be biologically plausible implantation of planning, consistent with behavioral data and hippocampal recordings [49].

Relatedly, we largely focus on the consequences of joint and independent clustering on goal generalization in our experimental design, and have not considered how joint and independent clustering make different predictions about how mappings are generalized. For example, if an independent clustering agent tries a response from the wrong mapping on the first trial of a new context, we would expect the agent to try to move in the same direction on the next trial using the corresponding response of a different mapping. We would not expect this tendency from a joint agent, because it would presumably use the observed mapping information to adjust its high-level plan. While our experiment was not designed to find this effect within-subjects, we can nonetheless look for this pattern of behavior in a cross experiment comparison. Using Bayesian logistic regression, we compared the tendency of subjects to "switch" from one mapping to another following an unsuccessful attempt to move, that is, move in the same direction as they would have in a previous trial had the previous mapping been valid for the context. We would expect this to be more common in experiment 2, as the task supports independent clustering, than in experiment 1, where the task supports joint clustering. While we find this prediction is directionally correct, we do not find strong statistical evidence for it ($\mathbb{E}[\beta_{\exp2>\exp1}] = 0.128$, one-tailed p-value = 0.03, HPD = [-0.015, 0.24]) in a Bayesian logistic regression that considers both training and test contexts and includes trial number, the number of trials within a context, whether a context was repeated sequentially and previously correct, and an individual subject bias term as nuisance regressors. One potential interpretation is that this follow-up analysis across experiments is simply underpowered. Further complicating this analysis is a difference in experience with the task in experiment 1 and experiment 2, which we attempt to control for statistically. As such, we can not draw strong conclusions from our data about how mapping generalization differentiates joint and independent clustering.

While we found consistent evidence for the most diagnostic qualitative patterns predicted by the meta-generalization agent across three experiments, not all of the predictions of the model were borne out. Our models of generalization, like previous clustering models of generalization, rely on the CRP prior. When used in context-popularity based clustering, the CRP predicts that contexts will be generalized as a parametric function of their popularity. While previous data confirm that subjects do have a strong bias to prefer more popular structures as a prior [4], that work did not assess whether such an effect was parametric (i.e., increasingly stronger preferences for increasingly more popular structures). We find mixed evidence for such a parametric effect in our data. In experiment 2, we found evidence that subjects generalize only the most popular goal. In experiment 3, two of the three significant test context contrasts relied on the distinction between the second and third most popular goal in training (specifically, the contrast between contexts 2 and 3 and the contrast between 1 and 4 vs. 2 and 3 involved these preferences). Assuming this partial null effect does not simply reflect an absence of statistical power, there are multiple potential explanations, the most simple of which is that experiment 2 was more difficult than experiment 3 (as it had more training contexts and more goals to choose from) and consequently, subjects had lower accuracy in experiment 2 than in experiment 3 (S5 Fig; experiment 2 vs. 3: difference score = -0.045, t = 2.3, p<0.03). Because subjects tend to extend less cognitive effort in more demanding tasks [50–52], it is possible that they relied on a less cognitively demanding strategy. Relatedly, the higher number of trials in the latter two experiments relative to the first may have encouraged subjects

to adopt a more cognitively demanding strategy, as they may have changed the relative costs and benefits of doing so.

The difference in task complexity (which extends to the first experiment as well) resulted from experimental choices designed to increase the interpretability of the experiments. Critically, the ability to resolve the predictions of joint and independent clustering is related to the complexity of the space of reward and mapping functions, as the models make different predictions only in explored regions of this space. We increased the space of contexts and goals in experiments 2 and 3 in order to differentiate model predictions on a within-subjects basis. In addition, we balanced the number of presentations of each context so that each goal would be correct the same number of times. Otherwise, goals associated with higher popularity contexts would have a higher expected value marginalized across contexts. These two choices interact such that contexts were experienced less frequently in experiment 2 than experiment 3, increasing the difficulty of the associative learning problem.

A further consequence of these choices is that working memory demands are not equal across the tasks. This consequently offers an alternative explanation to the meta-generalization agent: people may be arbitrating their generalization strategy based on the complexity of the task-space as opposed to reward prediction. How working memory would influence the arbitration strategy is not obvious, but intuitively, we might expect people to favor joint structure to the extent that working memory capacity supports it and then switch to independent structure as task complexity increases. Joint structure is more representationally greedy than independent structure, and such, we would expect it requires a higher memory load to learn the reward contingencies. Subjects learn reward contingencies more slowly under higher memory load [53] and we would expect working memory decay to affect joint and independent structure unequally. Joint clustering will necessarily result in at least as many context-clusters as independent clustering, meaning that the reward values of independent clustering are updated more frequently, possibly leading to better estimates of reward and thus, the meta-generalization agent may favor independent clustering as task-complexity increases. While we have not considered resource constraints in our model and they likely play an important role, it nevertheless remains clear from our data that people arbitrate between a compositional and non-compositional generalization strategy depending on task demands. Indeed, the same issue as to capacity limitations also arises in related literature on model-based vs. model-free contributions to learning [54], where highlighting this tradeoff across environments is nonetheless useful.

A related limitation is the auto-correlated presentation of training contexts in the tasks. While the approximate inference method used in the computational modeling (an approximate global MAP estimate) is generally insensitive to order manipulations, this does not appear to be true for humans subjects, who show substantial order effects in both learning [55] and in generalization tasks [8]. Moreover, we note that neural network model implementations of this context clustering have assumed a gating process in which prefrontal cortical neural populations are updating when the abstract structure changes [3]. Notably, the development and generalization of abstract structures in these gating networks is actually enhanced during block training [56], as in the empirical data; this amounts to non-exchangeable approximate inference [8]. However we have not explored this manipulation in the context of joint vs independent clustering. We would expect this pattern to extend here as well, but how the training regime interacts with meta-generalization is unclear.

Finally, in this study, we presented model predictions and behavioral analyses at a group level and do not make claims about individual subjects. This limitation is a consequence of the task design: we probe generalization in a small number of test trials chosen to differentiate the computational models qualitatively. There are only a few (2-4) trials per subject per measure

that distinguish the model predictions and most of the trials in each experiment are used to teach the subjects the statistics of the task or to balance the stimulus-action-outcome value of each goal and button-press. While we could attempt to fit a process model to each subject to determine each subject's generalization strategy, the estimated strategy of each subject will be highly dependent on a few trials and we are hesitant to rely upon this type of individual difference metric. As a result, it is difficult to make claims of whether individual subjects show a mixture of joint and independent clustering in experiment 3, or whether our findings are driven by two separate pools of subjects. We note that this distinction does not speak to whether individual subjects use a meta-generalization strategy within a given task, as a meta-generalization that deterministically chooses a strategy would produce two pools of subjects. At a group level, we would expect to see a negative relationship between the joint and independent effect if two separate pools of subjects drive the effects, a relationship we fail to see (see S5 Text).

More generally, the difficulty in identifying individual subject strategy is not a limitation of the grouped data because of its larger sample size. The models make unique and identifiable qualitative predictions on these tasks and the metrics of these predictions, the context contrasts, are orthogonal from each other and not correlated with overall performance. As such, we believe it is strong to make qualitative predictions about which conditions show better or worse performance especially when the predicted patterns are orthogonal as here.

## Materials and methods

### Ethics statement

All participants were compensated for their participation and gave informed consent as approved by the Human Research Protection Office of Brown University under protocol 0901992629, "How prefrontal cortex augments reinforcement learning."

### Subjects

We collected in subjects online using Amazon Mechanical Turk and psiTurk [57].

**Experiment 1.** 198 subjects completed the task, of which two subjects were excluded for reporting they took written notes during the experiment. Cluster analysis was used to asses subjects for non-performance. A Gaussian mixture model was fit to the two measures: trials following a correct trial of the same context and the overall accuracy in all other training context trials (see S1 Text). On the basis of this analysis, 67 subjects were excluded for non-performance.

**Experiment 2.** For experiment 2, we collected 153 subjects, five of which were excluded for reporting they took written notes. Cluster analysis was used to assess non-performance on the task. A Gaussian mixture model was fit to two measures: the accuracy in trials following a correct trial of the same context and the binomial chance probability in all other training context trials (see S1 Text). On this basis, we excluded 33 subjects from further analysis, leaving a total of 114 subjects.

**Experiment 3.** 151 subjects completed experiment 3, two of which were excluded for reporting they took written notes during the experiment. Using the same measures and analysis as in experiment 2, we used cluster analysis to exclude 34 of the remaining 149 subjects on the basis of their performance on these measures (see S1 Text).

## Task

In each of the three experiments, subject controlled agent in a 6x6 grid world which they had to navigate into a labeled ('A', 'B', 'C', or 'D') goal location. In each study, subjects had to learn both the value of the goals in each context as well as a "mapping" between button responses ('a', 's', 'd', 'f', 'j', 'k', 'l', and ';') and movement in the grid worlds (North, South, East, West). Each trial was a new instance of a grid world and associated with a color-cued context. To aide memory, contexts were autocorrelated in time. This was done by splitting the training contexts by permuting the order of the contexts under the constraints that the first half and second half of training were equally balanced and subject to a hazard rate of 25% in the first half of training that was lowered to 8% in the second half. The order of contexts was fully randomized in each test phase. In each context, one goal provided a fixed deterministic reward while all other goals were unrewarded. Separately, each context was also associated with one of two deterministic mappings. One mapping was always associated with left-hand keys on a standard US keyboard ('a', 's', 'd', 'f') and the other was always associated with the right-hand keys ('j', 'k', 'l', ';'). These two mappings were orthogonal for each hand, such that it was not possible to learn a mapping on one hand and transfer that knowledge directly to the other hand. Organized from left to right (i.e., 'a' to 'f' or 'j' to ';'), there were two possible mappings, either West, North, South, East, or North, West, East South. Each mapping was assigned to each uniquely across each hand, randomized across subjects, and each which mapping corresponded to the high or low popularity mapping was also varied across subjects.

It is worth noting these mappings are substantially more simple than the state-transition functions found in purely computational agents. This simplification was done to aid learning and the degree to which humans learn the transition structure was not an experimental question. As these mappings were deterministic and non-overlapping, each was identifiable with a single button press. Prior theoretical work [18] compared these reduced mappings to full state-transition functions and found that goal generalization was not affected by this simplification.

Subjects were required to learn both the identity of the correct goal and the mappings through exploration. To counterbalance for low-level action-values, the location of the goals and the starting point of the agent were randomized on each trial. In addition, barriers were randomly placed in a subset of trials to encourage additional planning. The relationship between keypresses and movements was permuted across subjects, as was the visible label attached to each goal.

Subjects were instructed that each context was paired with a single mapping and a single rewarded goal location and asked to use their left-hand and right-hand to navigate. Subjects were instructed that the shared color of the agent and goal cued the mapping and the value of the goals and that this relationship was constant across all trials with the same color. As an additional memory cue in experiment 2, each trial was labeled with a "room number" consistent with its context and all trials in a context within the experiment shared the same pattern of walls.

## Computational modeling

We used a version of the independent clustering, joint clustering and meta-generalization agent models adapted from [18] to generate qualitative predictions for each of the three experiments. Simulations were run generatively to inform the design of the experiments and make qualitative predictions. To verify the statistical significance of these predictions, we estimated the posterior distribution of the relevant comparisons via sampling. Specifically, we generated 2500 simulated samples of each model on each task and sub-sampled 200 batches of

simulations matched to the sample size in the human experiments (e.g. for experiment 2, we generated 200 batches of 115 simulations to match the 115 subjects collected). This allows us to estimate the probability that a randomly sampled experiment would produce the observed effects. Importantly, independent and joint clustering are matched in the number of free parameters and can be directly compared. Accuracy, or binary rewards collected per trial, are reported as opposed to total reward.

Each model was a reinforcement learning model that learned a mapping function between keyboard responses and movements in the maze as well as a value function over goals. Formally, we define a mapping function with a probability mass function $\phi(a, A)$, where $a$ defines a keyboard movement and $A$ is a movement in the maze. As rewards in the task are binary, the value function over goals can also be expressed as a probability distribution over goals $R(g) = \Pr(g)$. For all three agents, mappings and goals are estimated with maximum likelihood estimation. Agents were provided with fully supervised information as to the attempted movement direction in the case the agent attempted to move through a barrier, as this information was visually signaled to human participants as well.

The mapping and reward function was constant and deterministic for all trials within a context. The models differed in how they assumed these functions were generalized across contexts. Joint clustering assumes that each context $c$ belongs to a cluster of contexts $k$ that share a single mapping and reward function, $\phi_k$ and $R_k$. Generalization is then cast as the problem of inferring the correct assignment of $c$ into $k$ via Bayesian inference:

$$\Pr\left(c \in k | \mathcal{D}\right) \propto \Pr\left(\mathcal{D} | k\right) \Pr\left(c \in k\right) \tag{1}$$

where the likelihood function $\Pr\left(\mathcal{D} | k\right) = \prod_{\mathcal{D}} \phi_k(A|a) R_k(g)$ is the product of the observed probability of transitions and rewards on each trial for context cluster $k$. For the likelihood function, we use the maximum likelihood estimate over all trials. The prior is defined with a Chinese restaurant process [23], defined

$$\Pr\left(c \in k\right) = \begin{cases} \frac{N_k}{N+\alpha} \\ \\ \frac{\alpha}{N+\alpha} \end{cases} \tag{2}$$

where $N_k$ is the number of contexts previously assigned to a cluster, $N = \Sigma_k N_k$ is the total number of contexts visited thus far and $\alpha$ is a concentration parameter that governs the propensity for the process to assign a new context to a new cluster. For all of the simulations here, $\alpha$ was drawn from a standard log-normal distribution, such that $\log \alpha \sim \mathcal{N}(-0.5, 1)$, to simulate individual differences generalization. During action selection (discussed below) the *maximum a posteriori* (MAP) cluster assignment was used to approximate the posterior.

While joint clustering assumes mappings and goals generalize together, independent clustering loosens this assumption by assigning each context twice: once for mappings and once for goals. As before, cluster assignments are made via Bayesian inference (Eq 1) with the Chinese restaurant process prior (Eq 2) but with a different likelihood for mapping clusters and reward clusters. Mapping clusters use as their likelihood the mapping function $\Pr\left(\mathcal{D} | k\right) = \prod_{\mathcal{D}} \phi_k(A|a)$ whereas goal clusters use the reward function over goals as their likelihood $\Pr\left(\mathcal{D} | k\right) = \prod_{\mathcal{D}} R_k(g)$. While this can lead to a larger absolute number of clusters (with potentially two clusters per contexts vs. one per context in the joint model), independent clustering is a simpler statistical model as it does not represent co-occurrence statistics [18].

In contrast to both joint and independent clustering, the meta-generalization agent does not employ a fixed generalization strategy and instead dynamically arbitrates between joint and independent clustering. This is done via sampling, where each strategy is sampled

proportionally to how well it predicts reward in each novel context. Formally, each model is sampled according to its weight, $w_m$, which is defined as

$$w_m \propto \left( \prod_{t=1}^{n} \Pr\left(r_t | m\right) \right) \Pr\left(m\right) \tag{3}$$

where $m$ is either independent or joint clustering and $r_t$ is the reward observed on trial $t$. The prior over models, $\Pr(m)$, was chosen stochastically by sampling a uniform between zero and one. These model weights are an approximation of the unnormalized Bayesian model evidence and are consistent with a prediction error based arbitration strategy [18]. Thus, the meta-generalization agent favors joint or independent clustering to the degree to which it predicts unseen rewards.

Action selection was equivalent in all three models. Each agent had access to the structure of each grid-world on each trial in the form of a transition function $T(s, A, s')$. This transition function defined the probability of transitioning from location $s$ to location $s'$ having made the cardinal movement $A$. As movement in the grid-worlds was deterministic, this probability was always either one or zero. This was done to mirror human participants who have access to visual information indicating a spatial relationship between states as well as the locations of goals and the presence of barriers. We note that the spatial planning component of this task was intended to be simple for human participants.

This transition function was used to solve an action-value function on each trial in terms of cardinal movements,

$$Q(s, A | c) = \sum_{s'} T(s, A, s')[R_{\hat{k}}(s') + V_{\hat{k}}(s')] \tag{4}$$

where $R_{\hat{k}}(s')$ is the reward function over states for the MAP cluster assignment $\hat{k}$ and where $V_{\hat{k}}$ is the corresponding value function over states. Here, the reward function $R_{\hat{k}}(s')$ is expressed over locations in the grid-world instead of over goals. In the behavioral task, goal locations are randomized to account for stimulus-response biases and the location of each goal is provided to the agent as it is visually available to subjects.

The value function over states, which defines the discounted expected reward of each state under the optimal policy, is defined by the Bellman equation,

$$V_{\hat{k}}(s) = \max_{A} \left[ \sum_{s'} T(s, A, s')(R_{\hat{k}}(s') + \gamma V_{\hat{k}}(s')) \right] \tag{5}$$

where $\gamma$ is a discount parameter set to $\gamma = 0.8$ for all agents. This system of equations is solved using value iteration [17].

On each trial, a cardinal movement is sampled from a softmax function over the action-value function:

$$\Pr\left(A | s, c\right) \propto e^{\beta Q(s, A | c)} \tag{6}$$

where $\beta$, a free parameter, controls the propensity to choose the highest valued action. The value of $\beta$ was sampled for each simulation such that $\log \beta \sim \mathcal{N}(2.0, 0.5)$. The learned mapping functions were then used to sample keyboard responses,

$$\Pr\left(a | A\right) = \phi_{\hat{k}}(a | A) \tag{7}$$

As a comparison to the generalization agents, three contextual but non-generalizing agents were also simulated: a "Flat" agent, a Q-learning agent and an Upper Confidence Bound

(UCB) agent. The Flat agent is the most similar to the generalization agents, differing only in its assignments of contexts into clusters. The Flat agent assigns each context into a unique cluster, preventing the pooling of information between contexts. Conveniently, this can be interpreted as the limiting case of all of the generalizing agents with the concentration parameter $\alpha$ set to infinity and is thus otherwise the same.

The Q-learning agent further differed from the generalization agents in that the value of each goal was learned with a prediction-error based learning rule, defined:

$$R_c(g) \leftarrow R_c(g) + \eta(r - R_c(g)) \tag{8}$$

where $r$ is the observed reward, and $\eta \in [0, 1]$ is a learning rate. Like the Flat agent, the Q-learning agent learned both the mapping and the goals statistics of each context independently, so the reward function here is defined in terms of contexts and not clusters. For each simulation, a single fixed learning rate was sampled from the distribution $\text{logit}^{-1}(\eta) \sim \mathcal{N}(-1, 1)$, where $\text{logit}^{-1}(x) = 1/(1 + \exp(-x))$ is the inverse logit transform (thus the sampled value of the learning rate is bound between 0 and 1). Aside from this learning rule, the Q-learning agent was otherwise equivalent to the Flat agent.

Finally, the UCB agent incorporated estimates of uncertainty in the learned value of goals used during planning. By this, we mean that the value of each goal in a context was defined to be a function of the expected value of reward and the uncertainty of the estimate,

$$R_c(g) = \mu_{c,g} + \omega \sigma^2_{c,g} \tag{9}$$

where $\mu_{c,g}$ and $\sigma^2_{c,g}$ are the mean and variance of the rewards for goal $c$ in context $c$ and where $\omega$ is a free parameter that controls the degree of uncertain-guided exploration (see [58] for a thorough discussion of uncertainty guided exploration). In our simulations, $\omega$ was sampled from the distribution $\text{logit}^{-1}(\omega) \sim \mathcal{N}(-1, 1)$. Other than this estimate of reward, the UCB agent is equivalent to the Flat agent and Q-learning agent.

To estimate the mean and variance of rewards for the UCB agent, we used a time-varying normal approximation via a Kalman filter [59]. We define our Kalman filter with a series of update rules. The mean is updated according to the rule

$$\mu_{c,g} \leftarrow \mu_{c,g} + G(r - \mu_{c,g}) \tag{10}$$

and the variance by

$$\sigma^2_{c,g} \leftarrow (1 - G) \times (\sigma^2_{c,g} + \zeta) \tag{11}$$

where $\zeta$ is the diffusion noise that reflects the tendencies of reward to drift over time and where G refers to the "Kalman gain" (learning rate), defined

$$G = \frac{\mu_{c,g} + \zeta}{\mu_{c,g} + \zeta + \epsilon} \tag{12}$$

where $\epsilon$ is a form of irreducible noise. The values of $\zeta$ and $\epsilon$ were sampled from the distributions $\text{logit}^{-1}(\zeta) \sim \mathcal{N}(-1, 1)$ and $\text{logit}^{-1}(\epsilon) \sim \mathcal{N}(-1, 1)$, respectively. This estimate of mean and variance allows the estimate of reward to drift over time, with the consequence that the model will tend to over-explore the tasks defined here.

## Statistical analyses

**Analysis of computational models.** As noted above, we simulated 200 batches of simulations for each model matched to the human subject sample size for each condition in each

experiment. This represents a distribution on which we can directly calculate significance statistics and effect sizes. For each relevant comparison, we report the mean value (M), and the 95% highest posterior density interval (HPD) [60] and, when appropriate, the one-tailed p-test ($p_{\text{1-tail}}$) evaluated directly on this distribution. Statistical significance was determined by whether the measured effect included chance in its 95% HPD (chance was zero for the context contrasts). Because we evaluate 200 samples, the minimum p-value we report is $p_{\text{1-tail}} < 0.005$ for a single model. We do not report t-tests or other null-hypothesis significance tests on the computational models.

**Hierarchical Bayesian logistic regression.** We analyzed test context accuracy in each of the three experiments with hierarchical Bayesian logistic regression [60, 61], with a hierarchical prior over individual regression coefficients. All models were estimated using the No-U-turn sampler, a variant of Hamiltonian MCMC [62], implemented in the PyMC3 software library [63].

*A priori* model predictions were instantiated with and orthogonal set of contrasts between test contexts. Experiment 1 contained both within-subjects and between-subjects prediction and thus contained predictors for both. To control for within context learning, the number of trials in a context (*t*), whether a context was sequentially repeated and correct (rep) were included as nuisance predictors. It was modeled with the following logistic regression model:

$$\text{logit}(p) = \beta_s S + \beta_H H + \beta_t t + \beta_{rep}\text{rep} \tag{13}$$

where $S \in \{0, 1\}$ is equal to 1 for the (between-subjects) "Switch" condition, where "H"$\in\{0, 1\}$ is equal to 1 when the rewarded goal for the context is the high popularity goal. A hierarchical prior $\beta \sim \mathcal{N}(\mu_\beta, \sigma_\beta)$ was used for all predictors with a vague normal $\mu_\beta \sim \mathcal{N}(0, 100)$ and Cauchy hyper-prior $\sigma_\beta \sim$ Half- Cauchy(0, 100) for the mean and variance of each predictor, respectively.

The predictions of interest in experiments 2 and 3 were all within-subjects measures. As both experiments had 4 test contexts, a maximum of three orthogonal contrasts was possible. As before, the number of trials in a context, whether a context was repeated and previously correct were included as nuisance predictors. These were modeled as a linear combination of terms in a logistic regression,

$$\text{logit}(p) = \sum_c \beta_c c + \beta_t t + \beta_{rep}\text{rep} \tag{14}$$

where $\text{logit}(x) = 1/(1 + \exp(-x))$, *c* is a contrast, *t* is the trial number within a context and rep $\in\{0, 1\}$ indicates whether a context has been sequentially repeated. Hierarchical priors for the regression coefficients were defined in the same manner as experiment 1.

We report the expectation (mean value) of the regression coefficients (denoted as $\mathbb{E}[\beta]$). These were interpreted as group level effects and statistical significance was determined by gauging whether 0 fell within the 95% highest posterior density interval (HPD) [60] or using a 1-tailed test, denoted as $p_{\text{1-tail}}$, where appropriate. An advantage of this Bayesian approach is that we are able to define novel contrast with the same posterior sample through algebraic manipulation (i.e., addition or subtraction) of our parameters. For additional insight, we report relevant contrast created through this form of recombination.

**Analysis of goal selection.** In experiment 2, we analyzed the probability each goal was chosen in the first trial of a novel context at a group level with two Bernoulli models. For the first model, we collapsed the goal selection for all subjects across all test contexts, and modeled the goal-choice probability with independent Bernoulli distributions,

$$p_g \sim \text{Bernoulli}(\theta_g) \tag{15}$$

where $g \in \{A, B, C, D\}$ is a goal. In a separate analysis, we modeled goal choice probability as independent Bernoulli distributions, collapsing across all test contexts that shared a transition function. Thus, as test contexts 1 and 2 shared the same transition function, $\theta_g$ was modeled separately for contexts 1 and 2 then 3 and 4.

**Similarity analysis.** The qualitative predictions of the joint and independent clustering models were compared to human behavior across each of the three experiments. A vector of length two was created for each experiment for both human and model data. For human data, this consisted of the regression coefficients of the contrasts of interests and for the model data, this corresponded to the difference score between test conditions. Angle cosine was calculated between samples of these two sets of vectors as a metric of similarity, with a maximum similarity occurring when the angle cosine equals 1 and a measure of 0 representing orthogonal predictions.

## Supporting information

**S1 Fig. Cluster analysis of subjects training performance in experiments 1 (*A-D*) and 2(*E-H*).** *A,E*: Accuracy in repeated, correct trails vs. all other trials. *B,F*: Proportion of time the closest goal was selected by inclusion status. *C,G*: Subject rated difficulty by inclusion status. *D,H*: Time spend reading viewing instructions by inclusion status.
(TIF)

**S2 Fig. Experiment 1, training performance.** Accuracy (*A*), median reaction time (*B*) and the excess number of steps taken over the shortest path (*C*) shown as a function of the number of trials within each training context.
(TIF)

**S3 Fig. Experiment 2, training performance.** Accuracy (*A*), median reaction time (*B*) and the excess number of steps taken over the shortest path (*C*) shown as a function of the number of trials within each training context.
(TIF)

**S4 Fig. Experiment 3, training performance.** Accuracy (*A*), median reaction time (*B*) and the excess number of steps taken over the shortest path (*C*) shown as a function of the number of trials within each training context.
(TIF)

**S5 Fig. Cross experiment training accuracy.** *Left*: Accuracy as a function of the number of presentations in each context. Initial differences reflect a difference in chance accuracy between experiments *Right*: Accuracy as a function of number of presentations remaining (per context) within the training phase. Sharp drops in accuracy reflect the fact that each context was not shown the same number of times.
(TIF)

**S1 Table. Experiment 1 task design.** The number of trials within each context is balanced such that each goal and each mapping is presented the same number of trials across both training and test. Subjects saw either the "Repeat Test" context or the "Switch Test" contexts.
(PDF)

**S2 Table. Experiment 2 task design.** The number of trials within each context is balanced such that each goal and each mapping is presented the same number of trials across both training and test.
(PDF)

**S3 Table. Experiment 3 task design.** The number of trials within each context is balanced such that each goal and each mapping is presented the same number of trials across both training and test.
(PDF)

**S1 Text. Subject exclusion criteria.**
(PDF)

**S2 Text. Statistical analysis of training contexts.**
(PDF)

**S3 Text. Comparison between experiments.**
(PDF)

**S4 Text. Non-contextual agent.**
(PDF)

**S5 Text. Meta-generalization vs. mixture of subjects.**
(PDF)

## Acknowledgments

We thank Mark Ho for providing code used in the behavioral task and thank Matt Nassar for helpful discussions.

## Author Contributions

**Conceptualization:** Nicholas T. Franklin.

**Data curation:** Nicholas T. Franklin.

**Formal analysis:** Nicholas T. Franklin.

**Funding acquisition:** Michael J. Frank.

**Investigation:** Nicholas T. Franklin.

**Methodology:** Nicholas T. Franklin, Michael J. Frank.

**Project administration:** Nicholas T. Franklin, Michael J. Frank.

**Resources:** Michael J. Frank.

**Software:** Nicholas T. Franklin.

**Supervision:** Michael J. Frank.

**Visualization:** Nicholas T. Franklin.

**Writing – original draft:** Nicholas T. Franklin, Michael J. Frank.

**Writing – review & editing:** Nicholas T. Franklin, Michael J. Frank.

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
