## [Decision Letter · Decision Letter 0]

3 Dec 2019

Dear Dr Franklin,

Thank you very much for submitting your manuscript, 'Generalizing to generalize: humans flexibly switch between compositional and conjunctive structures during reinforcement learning', to PLOS Computational Biology. As with all papers submitted to the journal, yours was fully evaluated by the PLOS Computational Biology editorial team, and in this case, by independent peer reviewers. The reviewers appreciated the attention to an important topic but identified some aspects of the manuscript that should be improved.

We would therefore like to ask you to modify the manuscript according to the review recommendations before we can consider your manuscript for acceptance. Your revisions should address the specific points made by each reviewer and we encourage you to respond to particular issues Please note while forming your response, if your article is accepted, you may have the opportunity to make the peer review history publicly available. The record will include editor decision letters (with reviews) and your responses to reviewer comments. If eligible, we will contact you to opt in or out.raised.

- Supporting Information uploaded as separate files, titled 'Dataset', 'Figure', 'Table', 'Text', 'Protocol', 'Audio', or 'Video'.

We hope to receive your revised manuscript within the next 30 days. If you anticipate any delay in its return, we ask that you let us know the expected resubmission date by email at ploscompbiol@plos.org.

Sincerely,

Jill O'Reilly

Associate Editor

PLOS Computational Biology

Kim Blackwell

Deputy Editor

PLOS Computational Biology

[LINK]

Reviewer's Responses to Questions

**Comments to the Authors:**

Reviewer #1: Uploaded as attachment

Reviewer #2: The authors aim at investigating the computational processes that lead to generalization in human reinforcement learning. In particular, they contrast different models that implement joint, dis-joint and meta-generalization. There are many things to like about this paper. The tasks are very clever, the sample sizes are above average, the very elegant modeling and the important research question. I have reviewed this at another journal and I noted that in the present version of the manuscript, the authors addressed all the concerns I expressed in the first round of reviews (including additional simulations, better free parameters estimation and model-comparison) and also amended the discussion. I am happy to recommend for publication as is.

Reviewer #3: This paper investigates whether humans generalize task components in a conjunctive or independent manner as a function of the statistical structure of the task environment. The study describes three experiments in which participants were asked to navigate a grid world to a rewarded goal state. On each trial, participants had to identify one out of two valid mappings from keys to directions (mappings) to reach the relevant goal state (goals). The generalizable components of a task are operationalized as the correct mapping and the correct goal, respectively, and are both contingent on an observable context. Task structure is operationalized as the mutual information between mappings and goals, and varied across experiments. The study derives qualitative behavioral predictions for each experiment from three different non-generalizing agents, as well as three different generalizing agents described in earlier work (Franklin & Frank, 2018). The “joint” generalizing agent clusters contexts based on the joint statistics between goals and mappings. The “independent” generalizing agent assigns contexts to clusters that represent goals and mappings separately. The meta-generalization (“mixture”) agent arbitrates between joint and independent clustering. Computational results indicate that the optimal type of clustering (joint, independent, mixture) depends on the task structure, that is, the joint agent performs well in environments with high mutual information between goals and mappings whereas the independent agent performs well in environments with low mutual information between goals and mappings. The task structures deployed across the three experiments were found to arbitrate between the generalizing models. The manuscript concludes that behavioral results are overall consistent with a mixture of joint and independent clustering and that participants adapt their clustering strategy as a function of mutual information between task components.

The present manuscript nicely complements preceding work by contrasting human generalization performance against performance of three generalization models introduced by Franklin & Frank (2018). I believe that the scientific questions addressed by this study lie well within the scope of this journal and that the manuscript could be an interesting addition to the literature on human generalization and reinforcement learning. Moreover, I believe that the computational analyses and experimental setup are well suited to investigate how humans generalize task components as a function of statistical structure. I particularly commend the authors for their rigorous treatment of outlier detection and subject exclusion in their MTurk study. Overall, I found the manuscript to be well-written and enjoyed reading it.

The manuscript touched on most of my concerns in the Limitations section of the Discussion. However, remaining major concerns regard the generality of the study’s results with respect to blocked vs. interleaved training (Major Comment 1), a dissociation between mixed clustering within individuals vs. mixed clustering between individuals (Major Concern 2), the conclusiveness of difference scores (Major Comment 3) and the discussion of related work in machine learning (Major Comment 4). That said, I believe that all comments are addressable in a revision.

Below I provide a list of more detailed comments that will hopefully be useful in improving this manuscript.

Major Comments

1) Generality of joint clustering (blocked vs. interleaved training): Results from experiment 1 and 3 are taken as evidence that participants cluster task components jointly when there is high mutual information between task components. However, I am wondering whether this finding is contingent on the design choice that “contexts were autocorrelated in time” (p. 35). Recent experimental work indicates that such blocked training promotes the learning of factorized representations (e.g. for clusters of task components) whereas interleaved training impairs such acquisition (Flesch, 2018). However, if I understand correctly, then the joint agent, as well as the meta-generalization agent appear to be indifferent to this manipulation, that is, their clustering behavior should be independent of whether contexts are presented in a blocked or interleaved fashion. The reader may suspect that participants would show little to no evidence for joint clustering when task contexts are interleaved instead of blocked. This may be due to increases in memory demand, as touched on by the authors in the discussion. Thus, to conclude that participants show a general tendency for joint clustering when there is mutual information between task components, the manuscript would benefit from an additional version of experiment 1 with interleaved contexts in the training phase. Alternatively, the potential effects of blocked vs. interleaved training on structure learning could be treated in the Limitations section of the manuscript.

2) Meta-generalization vs. mixture of joint and independent agents in experiment 3: Results from experiment 3 are taken as evidence that participants’ behavior conforms best with the meta-generalization agent. However, the qualitative effects associated with the independent clustering strategy (contrast between test contexts 1 and 4), on the one hand, and the qualitative effects indicative of joint clustering (contrast between test contexts 2 and 3), on the other hand, may be driven by different subpools of participants. That is, instead of each participant adapting a mixture of independent and joint clustering (as suggested by the meta-generalization agent), the observed effects may be explained by a mixture of one set of participants adopting an independent clustering strategy (thus contributing to the observed contrast between test contexts 1 and 4) and a different set of participants adopting a joint clustering strategy (contributing to the observed contrast between test contexts 2 and 3). Behavioral analyses at the group level permit claims about individual subjects, as acknowledged by the authors (p. 34). However, dissociating these two hypotheses may not require fitting a model to each individual subject. Instead, one could statistically investigate whether evidence for the joint strategy (performance difference between test contexts 2 and 3) and the independent strategy (performance difference between test contexts 1 and 4) is driven by different subpools of participants.

3) Conclusiveness of difference scores: Admittedly, I found it difficult to draw conclusions about each agent’s behavior from the difference scores (e.g. Fig. 4). It might help to break down each agent’s performance as a function of experiment condition (e.g. accuracy of the joint vs. independent agent as a function of switch condition in experiment 1). For instance, to illustrate the benefit of the independent vs. the joint model in experiment 3, the reader might want to compare absolute performances between two models in test context 1 (I assume that the independent model is predicted to perform better than the joint model), as well as in test context 2 (here, I assume the joint model is predicted to perform better than the independent model), rather than just comparing performance between conditions within each agent and comparing overall performance between agents. Moreover, to provide further evidence of joint clustering in human participants, one would expect that participants would approach the goal predicted by the joint clustering model on the very first trial of a test context in the switch condition of experiment 1 (as soon as participants figure out the correct mapping). That is, on the first trial of the test context with a red mapping in the switch condition of experiment 1 (c.f. Fig. 3), the joint clustering model would predict that participants mistakenly approach goal A more than goal B, as the former was clustered with the red mapping in the training condition. If I am not mistaken, then this would be a testable prediction of the joint clustering model that seems worth an assessment over and above a performance difference between repeat and switch conditions across all trials.

4) Related work in machine learning: There are related approaches in machine learning targeting generalization across task environments, such as hierarchical reinforcement learning and clustering of action sequences (e.g. Thrun & Schwartz, 1995), zero-shot task generalization (e.g. Oh et al., 2017) or generalization through state space clustering (e.g. Hashemzadeh, Hosseini & Ahmadabadi, 2019; Mannor et al., 2004). Discussing the relationship to some of these ideas, and delineating the proposed agents from existing approaches would make this paper more accessible to the machine learning community.

Minor Comments

1) General: The manuscript would benefit from a proper definition of goal popularity and mapping popularity as experiment factors. Similarly, it might be worth adding a proper definition of the switch vs. repeat conditions to the statistical analysis section.

2) General: To facilitate interpretation of agent behavior for the reader, it would be helpful to depict each agent’s expected clustering in the form of Fig. 2 for each experiment.

3) P. 4-5: The intuition behind the tradeoff is still at bit unclear at this point. It might help to include an example.

4) Figure 2: The depiction of goal values and transitions on the right side appears to be redundant with Figure 1.

5) P. 12, “Each model was simulated with sampled parameters on 2500 random instantiations of the tasks”: I’m wondering why the models weren’t simulated on the exact same instantiations of the tasks that the human subjects performed to ensure proper comparison with human data?

6) P. 33: Model-based planning appears to be associated with a cognitive cost that participants arbitrate against the benefits of planning (Kool, Gershman, Cushman, 2017). Following this intuition, the discussion mentions memory demand as an alternative explanation to the meta-generalization agent. It might be worth to highlight in this context that a higher number of trials in the last two experiments (relative to experiment 1) could have led participants to adapt independent clustering as opposed to joint clustering (thus, explaining lesser evidence for joint clustering in experiments 2 and 3 compared to experiment 1).

7) P. 42: If I understand correctly, then the two nuisance factors (number of trials, context repetition) are supposed to capture learning within the test phase. However, human learning may not just be a function of context repetition, but whether participants reached the correct goal state in the previous trial. Therefore, the authors may want to consider including a nuisance factor coding for the interaction between context repetition and accuracy on the previous trial.

Typos

1) P. 4. “movement through and environment”  “movement through an environment”

2) P. 6: “such that is was not possible to learn”  “such that it was not possible to learn”.

3) P. 13: The text seems to incorrectly reference Fig. S3A (instead of Fig. S2A)

4) P. 29: “expect these tasks to produces”  “expect these tasks to produce”

5) P. 49: “and allowed for the and allowed for”  “and allowed for”

6) Experiment 3 appears to be missing a subsection heading “Computational Modeling”

References

Flesch, T., Balaguer, J., Dekker, R., Nili, H., & Summerfield, C. (2018). Comparing continual task learning in minds and machines. Proceedings of the National Academy of Sciences, 115(44), E10313-E10322.

Franklin, N. T., & Frank, M. J. (2018). Compositional clustering in task structure learning. PLoS computational biology, 14(4), e1006116.

Hashemzadeh, M., Hosseini, R., & Ahmadabadi, M. N. (2019). Clustering subspace generalization to obtain faster reinforcement learning. Evolving Systems, 1-15.

Kool, W., Gershman, S. J., & Cushman, F. A. (2017). Cost-benefit arbitration between multiple reinforcement-learning systems. Psychological science, 28(9), 1321-1333.

Mannor, S., Menache, I., Hoze, A., & Klein, U. (2004). Dynamic abstraction in reinforcement learning via clustering. In Proceedings of the twenty-first international conference on Machine learning (p. 71). ACM.

Oh, J., Singh, S., Lee, H., & Kohli, P. (2017). Zero-shot task generalization with multi-task deep reinforcement learning. In Proceedings of the 34th International Conference on Machine Learning-Volume 70 (pp. 2661-2670). JMLR.

**Have all data underlying the figures and results presented in the manuscript been provided?**

Reviewer #1: Yes

Reviewer #2: Yes

Reviewer #3: Yes

PLOS authors have the option to publish the peer review history of their article (what does this mean?). If published, this will include your full peer review and any attached files.

Reviewer #1: No

Reviewer #2: No

Reviewer #3: No

---

## [Decision Letter · Decision Letter 1]

11 Feb 2020

Dear Dr Franklin,

We are pleased to inform you that your manuscript 'Generalizing to generalize: humans flexibly switch between compositional and conjunctive structures during reinforcement learning' has been provisionally accepted for publication in PLOS Computational Biology.

Before your manuscript can be formally accepted you will need to complete some formatting changes, which you will receive in a follow up email. A member of our team will be in touch within two working days with a set of requests.

Best regards,

Jill O'Reilly

Associate Editor

PLOS Computational Biology

Kim Blackwell

Deputy Editor

PLOS Computational Biology

Reviewer's Responses to Questions

**Comments to the Authors:**

Reviewer #1: I thank the authors for carefully addressing my questions and concerns. I think the changes aide the understanding of the few aspects that were initially a bit unclear. As I was already mostly satisfied with the initial version of the manuscript, I am now happy to recommend publication.

Reviewer #3: I have reviewed the original version of this manuscript and was happy to see that the authors have addressed concerns raised by all reviewers with great care. I was particularly curious about the analysis of the first few button presses suggested by Reviewer 1 but I understand that there is not enough data to warrant conclusive inferences. All in all, I have no remaining major concerns, just a small clarification regarding minor comment 2): I was referring to the (colored) illustration of clustering solutions shown on the left side of Figure 2 which is easy to parse, and would help understand an agent’s cluster solution for each experiment. However, I leave the authors to decide whether they want to include additional illustrations in the final manuscript.

Potential typo (p. 3): “and a potential cost of”  “at a potential cost of”

**Have all data underlying the figures and results presented in the manuscript been provided?**

Reviewer #1: Yes

Reviewer #3: Yes

PLOS authors have the option to publish the peer review history of their article (what does this mean?). If published, this will include your full peer review and any attached files.

Reviewer #1: No

Reviewer #3: No

---

## [Editor Report · Acceptance letter]

31 Mar 2020

PCOMPBIOL-D-19-01651R1 

Generalizing to generalize: humans flexibly switch between compositional and conjunctive structures during reinforcement learning

Dear Dr Franklin,

I am pleased to inform you that your manuscript has been formally accepted for publication in PLOS Computational Biology. Your manuscript is now with our production department and you will be notified of the publication date in due course.

With kind regards,

Laura Mallard
